**METHOD**

# Unico: a unified model for cell-type resolution genomics from heterogeneous omics data

Zeyuan Johnson Chen[1,2†], Elior Rahmani[2*†] and Eran Halperin[2*]

†Zeyuan Johnson Chen and Elior Rahmani contributed equally.

*Correspondence:
eliorrahmani@mednet.ucla.edu;
ehalperin@cs.ucla.edu

[1] Department of Computer Science, University of California, Los Angeles, CA, USA
[2] Department of Computational Medicine, University of California, Los Angeles, CA, USA

**Abstract**

Most population-scale genomic datasets collected to date consist of "bulk" samples obtained from heterogeneous tissues, reflecting mixtures of different cell types. We introduce Unico, a *Uni*fied *cross-o*mics computational method designed to deconvolve standard two-dimensional bulk matrices (samples by features) into three-dimensional tensors (samples by features by cell types). Unico is the first principled model-based deconvolution method that is theoretically justified for any tissue-level genomic data. By deconvolving bulk gene expression and DNA methylation datasets, we demonstrate Unico's superior performance compared to existing methods, enhancing the ability to conduct powerful, large-scale genomic studies at cell-type resolution.

**Keywords:** Deconvolution, Decomposition, Cell-type specificity, Epigenomics, DNA methylation, RNA expression, Computational models, Statistical methods, Nonparametric models

## Background

Studying cell-type level genomic variation is critical for unveiling complex biology. Unfortunately, collecting large and well-powered datasets at cell-type resolution for population studies has yet to become common practice. Current single-cell datasets typically consist of data collected from no more than several dozens of individuals due to prohibitive costs; purifying cell types at scale using flow cytometry is laborious and often impractical, particularly for solid and frozen tissues for which cell isolation is very challenging [1–5].

Indeed, most transcriptomic and other genomic data types collected to date have been measured from heterogeneous tissues that consist of multiple cell types, resulting in vast amounts of large heterogeneous "bulk" genomic data (e.g., over two million bulk profiles publicly available on the Gene Expression Omnibus alone [6]). This rationalizes the development of computational methods that can disentangle the convolution of cell-type level signals that compose such bulk profiles. The premise, upon

successful implementation, offers a transformative capability to conduct powerful, large-scale studies at the cell-type level in multiple tissues and under numerous conditions for which large bulk data have already been collected.

Here, we propose a method for deconvolving 2-dimensional (2D) bulk data (samples by features) into its underlying 3-dimensional (3D) tensor (samples by features by cell types). Thus far, deconvolution methods have been tailored to specific data types [7–12]. In contrast, we introduce a *Uni*fied *cross-o*mics method, Unico, the first principled model-based deconvolution method that is theoretically applicable to any heterogeneous genomic data, enabled by an optimization scheme that makes minimal assumptions about the underlying data distribution. As we demonstrate through a comprehensive analysis of multiple gene expression and DNA methylation datasets, this generalization translates into superior performance over existing approaches and improves our ability to conduct powerful large-scale genomic studies at cell-type resolution.

## Results

### From bulk genomics to cell-type resolution: decomposition versus deconvolution

The study of bulk genomics routinely calls for *decomposition*, wherein an observed bulk data matrix is modeled as the product of two matrices: (i) cell-type proportions (fractions) of the samples in the data and (ii) per-feature cell-type genomic levels ("signatures"; Fig. 1a). This amounts to solving a matrix factorization problem. For a given bulk observation $x_{ij}$ of genomic feature $j$ in sample $i$, virtually all decomposition models share the following assumption:

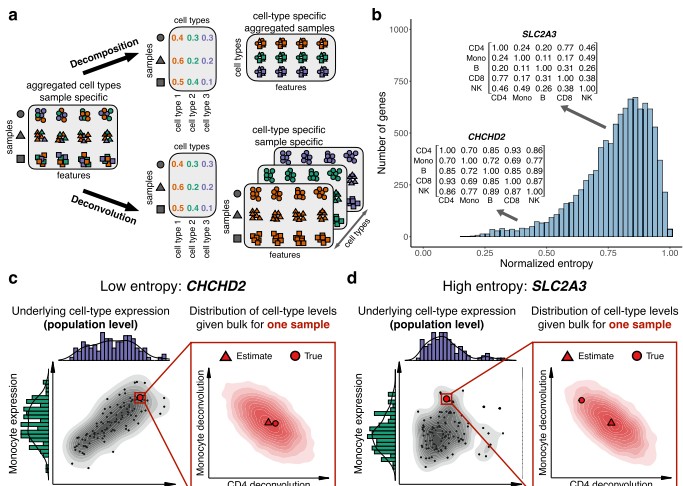

**Fig. 1** Overview of the Unico framework. **a** Illustration of decomposition versus deconvolution. **b** Cell-type covariance strength across the top 10,000 most highly expressed genes in scRNA-seq from PBMC [30], measured by normalized von Neumann entropy [31] (Methods). **c** The joint distribution of CD4 and monocyte expression in a low-entropy gene *CHCHD2* across 118 scRNA-seq PBMC samples [30] (left). For one arbitrary sample, Unico's estimated distribution of the possible CD4 and monocyte expression levels of the sample provides an accurate point estimate based on the sample's *CHCHD2* bulk expression (right). **d** Same as **c**, only for a high-entropy gene *SLC2A3*, providing less cell-type covariance information for the deconvolution

$$x_{ij} = \sum_{h=1}^{k} w_{ih} z_{jh} + e_{ij} \qquad (1)$$

where $w_{i1}, ..., w_{ik}$ are the proportions of $k$ modeled cell types in sample $i$, $z_{j1}, ..., z_{jk}$ are the cell-type level signatures of the genomic feature $j$ in each of the $k$ cell types, and $e_{ij}$ is an error term.

Numerous decomposition formulations with various assumptions on the products of the factorization have been proposed for the estimation of cell-type compositions and for learning cell-type signatures using different genomic modalities, including gene expression [13–16], DNA methylation [17–21], copy number aberrations [22, 23], ATAC-Seq [24], and Hi-C data [25]. The rich toolbox of decomposition methods has proven successful for a wide range of applications, such as clustering genes and studying their functional relationships [26, 27], inferring tumor composition [22, 23], and discovering cancer sub-types [28]. However, these methods allow us to infer only a single profile of cell-type level signatures per feature, which corresponds to the unrealistic assumption that all samples in the data share the same genomic levels at the cell-type level [29].

Every sample may reflect its own—possibly unique—cell-type level patterns, owing to various factors of inter-individual variation, such as genetic background, environmental exposures, and demographics. A natural adjustment of the decomposition model to reflect such variation yields:

$$x_{ij} = \sum_{h=1}^{k} w_{ih} z_{ijh} + e_{ij} \qquad (2)$$

where $z_{ijh}$ now represents the level of feature $j$ in cell-type $h$, specifically in sample $i$. Learning $z_{ijh}$ from bulk data is essentially a *deconvolution* problem, wherein we disentangle the mixture of signals in a 2D samples by features bulk data into the unobserved underlying 3D tensor of samples by features by cell types (Fig. 1a).

Decomposition under Eq. (1) can be viewed as a degenerate case of the more general deconvolution problem in Eq. (2) [29]. *Deconvolving* the data is thus more desired than merely *decomposing* the data, and the higher resolution of a successful deconvolution is expected to improve cell-type context and discovery in the analysis of bulk genomics. This has been highlighted and demonstrated by several recent deconvolution methods, including CIBERSORTx [8], CODEFACS [11], MIND [9], bMIND [10], and BayesPrism [12] in the context of transcriptomics and TCA [7] in the context of DNA methylation.

### Unico: a unified cross-omics deconvolution model

Current deconvolution methods can be categorized into two groups: heuristic approaches, including CIBERSORTx [8] and CODEFACS [11], and parametric methods, which assume the observed data follows a predefined distribution, including TCA [7], MIND [9], bMIND [10], and BayesPrism [12]. Parametric methods are susceptible to deviations from their underlying distributional assumptions. For instance, the common assumption that data follows a normal distribution is invalid for transcriptomic

data [32–34]. While variance-stabilizing transformations like log-scaling can empirically mitigate this issue in some cases, they also break the linearity assumptions in Eqs. (1)–(2), leading to biased estimates [35]. A more principled approach that accounts for the actual distribution of the data may yield more accurate and reliable results.

We introduce Unico, a deconvolution method for learning cell-type signals from an input of large heterogeneous bulk data and matching cell-type proportions. In practice, the latter is estimated from the input bulk profiles using reference-based decomposition (e.g., [15, 17]), as performed by all existing deconvolution methods [7–11]. The primary novelty of Unico stems from taking a model-based approach following Eq. (2) while making no distributional assumptions, which renders it the first principled model-based method that is theoretically justified for analyzing cell-type mixtures in any bulk genomic dataset (Methods).

A second key component of Unico is the consideration of covariance between cell types. Genomic features may be different yet coordinated across different cell types; for example, transcriptional programs can persist through multiple differentiation steps [36, 37]. Indeed, we observe that many genes present a non-trivial correlation structure across their cell-type-specific expression levels, as measured by entropy of the correlation matrix (Fig. 1b), with stronger cell-type correlations (lower entropy) observed between cell types that are close in the lineage differentiation tree (Additional file 2: Supplementary Notes). In the presence of covariance, Unico leverages the information coming from the coordination between cell types to improve deconvolution (Fig. 1c, d).

### Establishing a new state-of-the-art deconvolution for bulk genomics

We compared Unico to CIBERSORTx, TCA, bMIND, and BayesPrism, as well as to a simple baseline approach of naively weighting each bulk profile by the cell-type proportions of the sample. Our evaluation excluded methods that are either not publicly available [11] or require multiple measurements for every sample [9].

In order to form a basis for evaluation, we generated pseudo-bulk mixtures using single-cell RNA sequencing (scRNA-seq) data from peripheral blood mononuclear cells (PBMC; $n = 118$ donors) [30] and from lung parenchyma tissues ($n = 90$ donors) [38] (Methods). We first evaluated the performance of Unico and the other methods that do not incorporate cell-type profiles as prior information, including the baseline deconvolution, CIBERSORTx, TCA, and bMIND (without prior). Most methods estimated population-level means and variances similarly well, based on gold standard estimates using the known underlying cell-type profiles of the mixtures (Fig. 2a; Additional file 1: Fig. S1). Unico stands out as the leading method for learning cell-type level covariances, showcasing an average correlation improvement of 36.3% over bMIND, the second-best performing method, which also explicitly models cell-type covariances [10] (Fig. 2a; Additional file 1: Fig. S1).

Next, we evaluated how well the 3D tensor estimated by Unico correlates with the true underlying cell-type expression levels of the pseudo-bulk profiles. Unico consistently outperformed the alternative methods across all cell types, providing an average improvement of 17.8% in correlation over TCA, the second-best performing method (Fig. 2b; Additional file 1: Fig. S1). Unlike Unico, bMIND and BayesPrism are Bayesian deconvolution methods that can incorporate prior information on the distribution of

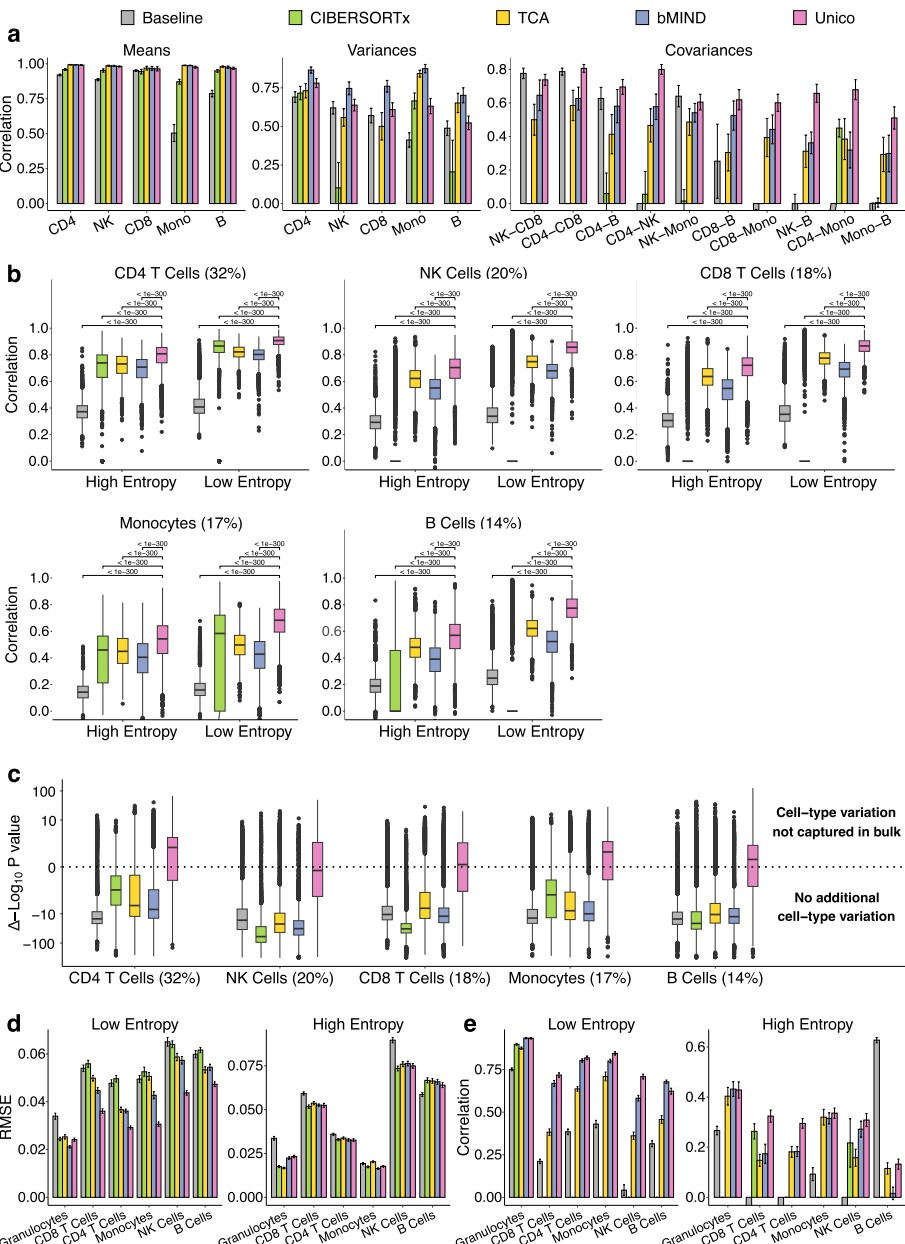

**Fig. 2** Evaluation of deconvolution methods. **a** Correlation between deconvolution and single-cell-based estimates of population-level means, variances, and covariances at the cell-type level across 20 sets of pseudo-bulk mixtures from PBMC scRNA-seq profiles of five cell types (500 samples and 600 randomly selected genes in each set). **b** Evaluation of the concordance between the deconvolution estimates and the known cell-type profiles of the same data in **a**. Boxplots reflect the distribution of linear correlation across all genes, and percentages indicate average cell-type abundances. **c** Assessing deconvolution estimates for their information that cannot be explained by pseudo bulk expression. Boxplots reflect the distribution across genes from the same data in **b** of $\Delta \log_{10}(p\text{-value})$, the difference between the log-scaled $p$-values of the effects of the pseudo bulk expression and those of the deconvolution estimates (higher is better; Methods). **d**, **e**, Evaluation of whole-blood DNA methylation deconvolution in terms of RMSE and correlation between estimates and experimentally validated cell-type level methylation across 20 random sets of 1000 highly variable CpGs. All barplots and error bars in the figure represent means and one standard deviation errors; negative correlations were truncated for visualization purposes, and $p$-values were calculated using a paired Wilcoxon test

the cell-type-level patterns. We found that prior information did not allow bMIND and BayesPrism to perform better than Unico. Remarkably, this remained true even in the best-case scenario, where the prior was learned from the true cell-type levels of all samples in the data (Methods; Additional file 1: Figs. S2 and S3).

As anticipated, the improvement of Unico is more pronounced in genes that exhibit strong cell-type covariance structure (low-entropy genes; average correlation improvement of 20.0%) compared to high-entropy genes (average improvement of 14.9%). This discrepancy highlights the added information Unico gains by modeling the cell-type covariance structure. Importantly, learning a richer model does not come at the cost of significant computational runtime in this case; in fact, Unico is the second fastest deconvolution method (Additional file 1: Fig. S4). The overall ranking of methods remained consistent across different numbers of modeled cell types and various sample sizes (Additional file 1: Figs. S5–S10), as well as across varying levels of noise added to the cell-type proportions input (Additional file 1: Figs. S11 and S12; Additional file 2: Supplementary Notes). Comparing Unico across increasing sample sizes demonstrates an expected increase in correlation between the deconvolution estimates and the true underlying cell-type-specific expression levels, with improvement plateauing above 250 samples (Additional file 1: Figs. S13 and S14).

Next, we assessed how well Unico and the other deconvolution methods can capture true cell-type profiles relative to bulk expression. Although this comparison is not strictly equivalent—since bulk profiles do not resolve cell-type–specific levels—it provides a useful benchmark. In particular, bulk expression is expected to correlate strongly with cell-type profiles for genes exhibiting similar expression across all cell types. Strikingly, in our benchmarking, Unico was the only method that produced estimates more strongly correlated with the true underlying cell-type profiles than both the corresponding pseudo-bulk levels and the pseudo-bulk levels with cell-type composition regressed out (Additional file 1: Figs. S15–S18). To further quantify this advantage, we applied a statistical model that measured how much additional variation each method explained in the true cell-type profiles beyond what was already captured by pseudo-bulk (Methods). This analysis confirmed that Unico was the only method that recovered significant additional variation across most genes, including in low-abundance cell types (Fig. 2c; Additional file 1: Figs. S1–S3 and S5–S12). Evaluating this metric across varying sample sizes further showed that Unico's advantage becomes robust for datasets exceeding 100 samples (Additional file 1: Figs. S13 and S14).

Lastly, we aimed to confirm that Unico can be utilized for other data modalities by deconvolving bulk DNA methylation data. Reinius et al. [39] assayed whole-blood and sorted cell methylation (six immune cell types) from the same six individuals. These experimentally measured profiles established a ground truth for the cell-type levels composing the whole-blood bulk samples. In order to circumvent the sample size limitation of the Reinius data ($n = 6$), we took a two-step, reference-based approach. Initially, we employed Unico to estimate the model parameters using a separate large whole-blood methylation dataset from a similar population [40]. Subsequently, we utilized these parameter estimates in Unico's tensor estimator. A similar procedure was adapted for the competing methods (Methods).

Unico demonstrated exceptional performance compared to the alternative methods in reconstructing the experimentally known 3D tensor. Considering the top 10,000 most variable methylation CpGs in the data, Unico achieved an average improvement of 8.8% and 8.1% in root median squared error (RMSE) and correlation compared with bMIND, the second best performing method (Fig. 2d,e; Additional file 1: Figs. S19 and S20). Incorporating the ground truth cell-type profiles as prior information, reflecting the best-case scenario, allowed BayesPrism (but not bMIND) to perform better than Unico; however, incorporating the same information in Unico's deconvolution (Methods) outperformed BayesPrism's average correlation by 15.7% (Additional file 1: Fig. S21). The ranking of the methods was preserved when considering a set of 10,000 randomly selected CpGs; unsurprisingly, all methods present a noticeable decrease in performance in this case (Additional file 1: Figs. S22–S25).

### Detecting cell-type-specific differential expression in heterogeneous tumors

Follicular lymphoma (FL) is the second most common indolent non-Hodgkin lymphoma (NHL) in the USA and Europe, accounting for nearly 20% of all NHL cases [41]. Previous work using FACS-sorted B cells from FL tumors identified 612 differentially expressed genes in the presence of *CREBBP* mutation [42]. Here, similarly to previous analysis [8], we asked whether deconvolving bulk FL tumors ($n = 24$, including 14 with *CREBBP* mutation) [8, 42] would allow us to detect the previously reported effects in B cells from FL tumors. We applied different deconvolution methods to estimate B cell expression levels from the bulk FL tumors, and we evaluated if they could recapitulate previously reported down- and up-regulation effects, identified from sorted B cells (gold standard). Unico is the only method consistently ranked as the top-performing method in its concordance with the gold standard for both the down- and up-regulated set of genes (Fig. 3a). In the down-regulated genes, Unico performed on par with bMIND with single-cell FL data as prior [43]. Both methods performed better than all alternatives, and remarkably, they were the only deconvolution methods that performed significantly better than a straightforward bulk analysis (adjusted $p$-value$< 3.28e{-}08$; paired Wilcoxon test). In the up-regulated genes, Unico and most other methods performed similarly well, except for the baseline method, bMIND and BayesPrism with the single-cell data as prior.

### Unico improves resolution and robustness in epigenome-wide association studies

We expected that modeling and effectively estimating cell-type covariance would allow Unico to yield better performance in downstream applications that aim at disentangling signals between cell types. In order to demonstrate this, we evaluated the different deconvolution methods in calling cell-type level differential methylation (DM). While ground truth DM is generally unknown, one can consider the consistency of a given method across different datasets as a surrogate for true/false positive/negative rates.

We applied each method for testing a set of 177,207 CpGs for cell-type level DM in four large whole-blood methylation datasets ($n > 590$ each) with sex and age information [40, 44, 45]. Specifically, for every possible combination of two out of the four datasets as discovery and validation data, we measured the consistency between datasets using the Matthews correlation coefficient (MCC) [46] (Methods). We excluded from

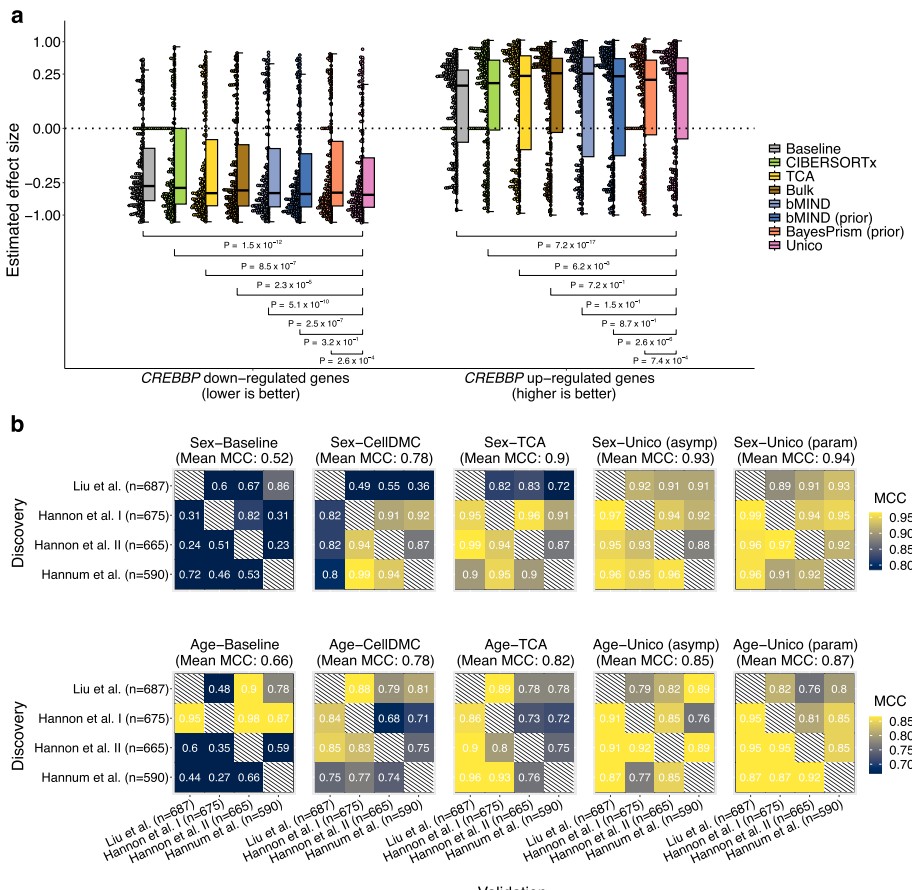

**Fig. 3** Application of deconvolution to downstream analysis tasks. **a** Deconvolution of bulk FL tumor samples for assessing previously reported *CREBBP* mutation-related gene expression in B cells. Presented are deconvolution-based B cell effect size distributions for 219 down-regulated and 275 up-regulated genes; comparisons to Unico were calculated using a one-sided paired Wilcoxon test. "(prior)" indicates incorporating prior cell-type level information derived from single-cell data [43]. **b** Consistency in calling cell-type level differential methylation with sex and age across four independent whole-blood DNA methylation datasets. Color gradients represent the Matthews correlation coefficient (MCC) for every possible pairing of two datasets as discovery and validation (Methods). "Unico (asymp)" performs a distribution-free statistical testing (asymptotic *p*-values); "Unico (param)" makes a parametric assumption that methylation levels follow a normal distribution

this analysis CIBERSORTx, due to its runtime (Additional file 1: Fig. S4) and poor performance in deconvolving bulk methylation (Fig. 2e; Additional file 1: Figs. S19–S25). Instead, we considered CellDMC, a method designed specifically for detecting cell-type level DM by evaluating linear effects of interaction terms between the condition of interest and cell-type proportions [47].

We first evaluated the methods that do not require prior knowledge of cell-type profiles. Unico outperformed the alternative methods, showing the highest overall consistency in detecting cell-type-level DM with sex and age across the four datasets (Fig. 3b), while also achieving calibration under the null (Additional file 1: Figs. S26 and S27). Despite incorporating prior information from experimentally sorted cell-type profiles [39], bMIND and BayesPrism, significantly under-performed compared to Unico and methods specifically tailored for DM (Additional file 1: Fig. S28).

We then investigated which factor contributes most to Unico's improved performance in consistently detecting DM: its distribution-free (i.e., non-parametric) approach or its ability to model and accurately estimate cell-type covariances. To isolate the source of improvement, we also created a parametric version of Unico. This version makes an additional assumption, used by all the methods tailored for DM, that methylation levels follow a normal distribution (Methods). The parametric and non-parametric versions of Unico produced highly concordant results (Additional file 1: Figs. S29 and S30), with only a slight improvement in favor of the parametric approach (Fig. 3b). These findings suggest that most of Unico's advantage over TCA, the second-best method in calling for cell-type DM, can be attributed to modeling cell-type covariances.

The above evaluation disregards a straightforward analysis of the bulk data, which cannot associate DM with specific cell types but rather call CpGs as generally associated with conditions ("tissue-level" analysis). Intuitively, models that provide cell-type resolution are more realistic and are thus expected to improve cross-dataset consistency over a standard tissue-level analysis. In order to verify this intuition, we evaluated a standard linear regression analysis of the bulk data for calling tissue-level DM (Additional file 1: Fig. S31). We observe that cell-type level analysis using any of the deconvolution methods provides a substantial improvement in consistency compared to the bulk analysis. In particular, Unico provides an increase of 107.5% and 40.7% in MCC for sex and age, respectively. Further adapting the different deconvolution methods to call tissue-level DM (Additional file 2: Supplementary Notes) yields all methods as better than a standard bulk analysis, with Unico being the top performing method (Additional file 1: Fig. S31). These results demonstrate how carefully modeling the cell-type signals in bulk data improves analysis even if constrained to a tissue-level context.

## Discussion

We propose Unico, a deconvolution method that is theoretically appropriate for any bulk genomic data type that reflects mixtures of signals across cell types. Here, we demonstrate the utility of Unico for gene expression and DNA methylation, however, our distribution-free treatment suggests its applicability to other genomic data types as well. Unico leverages covariance across cell types, and as such, it deconvolves particularly well low-entropy features that exhibit non-trivial correlation structures between cell types. Our evaluation, based on two scRNA-seq datasets from different tissues and purified methylation data, demonstrates that Unico considerably outperforms state-of-the-art methods in general, even when deconvolving high-entropy features.

The premise of deconvolution is that we can advance insights beyond what standard bulk analysis can provide by modeling cell-type patterns in relation to biological conditions. The correlation of deconvolution estimates with cell-type profiles is therefore an indirect, yet common, way to assess the performance and potential utility of these methods. Moreover, directly using deconvolution estimates in downstream analyses is often suboptimal: they reflect point estimates that are inherently noisy and represent only the "best guess" of a given model. Instead, deconvolution models can be designed to account for estimation noise in specific downstream tasks. For instance, similar to TCA and bMIND, Unico estimates the distribution of the underlying cell-type levels, providing not only point estimates but also estimates of their variance (i.e., uncertainty). In our cell-type differential

analysis with Unico, we incorporated this uncertainty into downstream statistical models in a one-step approach, thereby avoiding reliance on explicit deconvolution point estimates. By down-weighting cell-type profiles with particularly high estimated variance, we leveraged information that would otherwise be lost when relying solely on point estimates. We believe that future efforts in deconvolution should similarly design models that are tailored to specific downstream tasks and move beyond explicit point estimates.

Testing for differential cell-type variation within our framework models the effect of covariates and phenotypes on cell-type genomic variation. Unico can also be applied to model the opposite direction (i.e., the effect of cell-type genomic variation on a phenotype). However, unlike TCA and bMIND, this currently requires using point estimates of the cell-type patterns as the explanatory variables within a regression framework. Given recent findings that highlight how properly specifying the direction of the model can significantly influence the statistical properties of hypothesis testing [29], future efforts could potentially enhance power and calibration by extending Unico to learn the opposite direction without relying on explicit point estimates.

Finally, Unico has some limitations, and while some are not unique to Unico but are rather common to all the deconvolution methods we evaluated, they may potentially bias and affect the performance of our proposed model. First, Unico makes the assumption that cell-type proportions of the input bulk samples are known. Admittedly, this information is rarely available in bulk genomics data, so proportions need to be estimated in practice. While it is commonplace to employ reference-based methods for learning cell-type compositions, using estimates in place of measurements creates a source of noise and potential bias. Our multiple real data analyses using reference-based estimates, as well as our evaluation of deconvolution of in-silico mixtures under noisy cell-type compositions, suggest that, in practice, Unico is overall robust to this source of noise. Yet, all methods, including Unico, are expected to perform poorly when attempting to model a large number of cell types, as lowly abundant cell types represent only a small fraction of the variation in bulk data. Since heterogeneous tissues often represent mixtures of a large number of cell types and subtypes, the deconvolution of Unico may be biased by unmodeled cell types.

Another limitation of Unico pertains to its distribution-free approach. By not adhering to a pre-specified distribution, Unico is theoretically well-suited for various data modalities. However, this flexibility makes it inherently more sensitive to outliers. Unlike parametric methods, which tend to down-weight outliers due to their reliance on specific distributional assumptions, a distribution-free approach like Unico, which aims to capture any underlying distribution, may allow outliers to exert a disproportionate influence on the statistical inference. We carefully addressed this by excluding outliers during the parameter estimation and statistical testing (Additional file 2: Supplementary Notes). Failing to properly remove outliers could lead to sub-optimal results (e.g., calling cell-type-level DM less consistently across datasets; Additional file 1: Fig. S32)

## Conclusions

Unico deconvolves bulk genomic data into cell-type-level profiles, outperforming existing methods across gene expression and DNA methylation datasets. Using a distribution-free approach and modeling cell-type covariance, Unico provides a more precise

analysis of bulk data and enhances the ability to conduct powerful, large-scale genomic studies at cell-type resolution.

## Methods

### Unico: a model for uniform cross-omics deconvolution

We denote $X_{ij}$ to be the (tissue-level) bulk gene expression of gene $j \in \{1..., m\}$ in sample $i \in \{1..., n\}$. Hereafter, we use the notion of gene expression, however, $j$ can represent any other genomic feature that may vary across cell types. For simplicity of exposition, modeling of covariates with potential cell-type–specific effects and tissue-level covariates, such as batch effects, is deferred to the Additional file 2: Supplementary Notes. For the core Unico model, we assume:

$$X_{ij} = w_i^T Z_{ij} + e_{ij} \tag{3}$$

$$\mathrm{E}[e_{ij}] = 0, \mathrm{Var}[e_{ij}] = \tau_j^2 \tag{4}$$

The first term in Eq. (3) defines $X_{ij} \in \mathbb{R}$ as a weighted linear combination of cell-type expression levels. Specifically, $w_i = (w_{i1}, ..., w_{ik})^T \in \mathbb{R}^k$ is a vector of sample-specific cell-type proportions of $k$ cell types that are assumed to compose the studied tissue, and $Z_{ij} = (Z_{ij1}, ..., Z_{ijk})^T \in \mathbb{R}^k$ is a vector of the $k$ cell-type expression levels of gene $j$ in sample $i$. The second term $e_{ij} \in \mathbb{R}$ in Eq. (3) is an i.i.d. component of non-systematic variation with variance $\tau_j^2 \in \mathbb{R}$, reflecting measurement noise. We assume the cell-type proportions are fixed and given. In practice, these can be estimated using a reference-based approach (e.g., [15, 17]), as suggested by other deconvolution methods [7–11]). In contrast to a standard decomposition problem, which assumes shared cell-type expression levels across all samples, the unknown $Z_{ij}$ components are modeled as random vectors; this is emphasized by the use of upper-case notation. Specifically, for $Z_{ij}$, we assume:

$$Z_{ij} = \mu_j + \epsilon_{ij} \tag{5}$$

$$\mathrm{E}[\epsilon_{ij}] = \vec{0}, \mathrm{Var}[\epsilon_{ij}] = \Sigma_j \tag{6}$$

where $\mu_j = (\mu_{j1}, ..., \mu_{jk})^T \in \mathbb{R}^k$ is a vector of the mean cell-type levels, specific to gene $j$, and $\epsilon_{ij} = (\epsilon_{ij1}, ..., \epsilon_{ijk})^T \in \mathbb{R}^k$ is a noise term, i.i.d. with respect to samples, with mean zero and symmetric positive semidefinite variance-covariance matrix $\Sigma_j \in \mathbb{R}^{k \times k}$, which reflects the covariance of gene $j$ across cell types.

The Unico model makes no assumptions on the distribution of the components of variation in Eqs. (3)–(6), which makes it naturally applicable to all heterogeneous tissue-level omics that can be represented as linear combinations of cell-type-level signals. Finally, Unico can be viewed as a generalization of the TCA model and as a frequentist alternative for the bMIND model. See Additional file 2: Supplementary Notes for details.

*Estimating the underlying 3D tensor with Unico*   Given a single realization $x_{ij}$ of the bulk level coming from $X_{ij}$, we wish to learn $z_{ij}$, the realization of the cell-type-specific expression levels $Z_{ij}$ of the corresponding sample $i$ and gene $j$. Our goal is hence to

compose a 3D tensor (samples by genes by cell types) based on the 2D input matrix. We address this problem by setting the estimator of $z_{ij}$ to be the expected value of the conditional distribution $Z_{ij}|\theta_j, X_{ij}, w_i$:

$$\hat{z}_{ij} = \mathrm{E}\left[Z_{ij}|\theta_j, X_{ij} = x_{ij}, w_i\right] \tag{7}$$

where $\theta_j$ is the set of parameters that are specific to gene $j$, that is,

$$\theta_j = \{\mu_j, \Sigma_j, \tau_j^2\} \tag{8}$$

The following theorem provides an analytical solution for the 3D tensor estimator $\hat{z}_{ij}$, given the model parameters $\theta_j$ and model input: the bulk profiles $x_{ij}$ and the corresponding cell-type proportions $w_i$. In practice, as mentioned above, cell-type proportions estimated using external decomposition methods are provided as part of the input, from which we subsequently estimate the model parameters, as described later.

### Theorem 1

(*The Unico 3D tensor estimator*)

*For normally distributed bulk levels, the solution for the estimator stated in Eq.* (7) *under the Unico model is given by*:

$$\hat{z}_{ij} = \mathrm{E}[Z_{ij}|\theta_j] + \frac{\Sigma_{jw_i}\left(x_{ij} - w_i^T \mu_j\right)}{Sum\left((w_i w_i^T) \odot \Sigma_j\right) + \tau_j^2}$$

*where the $\odot$ operator is the Hadamard product of two matrices, and the* $\mathrm{Sum}(\cdot)$ *operator is a summation across all entries of a matrix.*

Proof is given in the Additional file 2: Supplementary Notes. The assumption that bulk levels follow a normal distribution limits the generalizability of this theoretical result. However, it allows us to analytically derive an efficient estimator. Importantly, our benchmarking results suggest that empirically, this estimator provides a good approximation even under empirical violations of the assumption (e.g., in deconvolving gene expression levels). Furthermore, this assumption does not affect the distribution-free estimation of the model parameters, which we describe in the following subsection.

Unico essentially defines the estimator $\hat{z}_{ij}$ as the expected value of the conditional distribution $Z_{ij}|\theta_j, X_{ij} = x_{ij}, w_i$, which aims to distribute the residual between observed and the expected bulk level back to individual cell types along directions specified by $\Sigma_j w_i$. This general framework was previously suggested in TCA [7]. However, under the richer Unico model, this conditional distribution becomes more informative owing to the correlation structure between cell types (i.e., $\Sigma_j$), which is not modeled and assumed to be the identity matrix in TCA. Intuitively, learning cell-type levels that better capture cell-type covariance will enhance our capacity to assign deconvolution signals accurately to the respective cell types in downstream analysis.

A priori, one may wonder whether modeling cell-type covariance is necessary for a deconvolution method to recapitulate the true cell-type covariance in the data. Put

differently, one could expect an accurate deconvolution method to capture cell-type covariance regardless of an explicit modeling of the covariance. However, our empirical results suggest that such modeling is valuable, and the following theorem provides intuition into why modeling the covariance is indeed desired in order to achieve accurate deconvolution. Besides Unico, TCA [7] is the only existing deconvolution method that offers an analytical estimator for the 3D tensor. Hence, the following exclusively focuses on Unico and TCA, as the theoretical analysis for other methods remains unclear.

### Theorem 2

(*Improved capacity to reduce covariance bias*)

*Assume for simplicity*: $\mu_j = 0$, $\Sigma_j = I_k$, $\tau_j^2 = 0$. *If $n \to \infty$ then* (*i*) *the cell-type covariances of the* 3D *tensor estimated by TCA are fixed and do not depend on feature j, and* (*ii*) *the cell-type covariances of the* 3D *tensor estimated by Unico are a function of the cell-type covariance of feature j.*

Proof is given in the Additional file 2: Supplementary Notes.

### Estimation and optimization

Given the bulk profiles $\{x_{ij}\}$, cell-type proportions $\{w_i\}$, we estimate the parameters $\{\theta_j\}$ of the model per feature *j* by following concepts from the Generalized Method of Moments (GMM) [48, 49]. The GMM framework allows us to learn the parameters of a model by iteratively solving Equations (*moment conditions*) that match population moments (or, more generally, a function of population moments) with their corresponding data-derived sample moments. While traditional GMM assumes a shared distribution across all samples, the Unico framework models each sample $X_{ij}$ as following a different distribution, governed by the sample-specific cell-type proportions $w_i$ and the parameters $\theta_j$. This leads to the formulation of sample-specific moment conditions $\{f_i(\theta_j, X_{ij})\}_{i=1}^{n}$, defined over a shared feature-specific set of parameters $\theta_j$. For example, we construct the first moment condition to match the expectation of $X_{ij}$ ($E[X_{ij}] = w_i^T \mu_j$), a function of $\mu_j$, with the observation $x_{ij}$, requiring their difference to be 0 in expectation. Specifically,

$$f_i(\theta_j, X_{ij}) = f_i(\mu_j, X_{ij}) = E[X_{ij}] - x_{ij} \tag{9}$$

$$E[f_i(\theta_j, X_{ij})] = 0 \tag{10}$$

Similarly for the second moment, we require:

$$E[f_i(\Sigma_j, \tau_j^2, X_{ij})] = E[\text{Var}[X_{ij}] - (x_{ij} - E[X_{ij}])^2] = 0. \tag{11}$$

Since the number of moment conditions *n* far exceeds the number of parameters $|\theta_j|$, the system is overdetermined and there is no solution that satisfies all equations. We thus seek to minimize the following:

$$\hat{\Theta} = \underset{\Theta}{\text{argmin}}\, f(\Theta, X)^T \hat{U} f(\Theta, X) \tag{12}$$

where $f(\Theta, X) = f_1(\theta_j, X_{1j}), ..., f_n(\theta_j, X_{nj})$ is the set of $n$ moment conditions, one per sample, and $\hat{U} \in \mathbb{R}^{n \times n}$ is a positive definite weight matrix, reflecting the inverse of the empirical variance-covariance matrix of the moment conditions, as in a typical GMM framework.

Assuming independence between samples implies independence between moment conditions $f_i(\theta_j, X_{ij})$ and simplifies $\hat{U}$ to be diagonal. Thus, it reduces the problem to a weighted least-squares formulation, resulting in an estimator that is asymptotically consistent and efficient. In practice, however, due to finite data, we choose $\hat{U}$ to only down-weight samples whose predictions incur large penalties on the moment conditions. Additionally, we incorporate informative constraints tailored to biological data (e.g., non-negativity constraints on the population-level cell-type means). Full details of the optimization and implementation are provided in the Additional file 2: Supplementary Notes.

### Implementation of Unico and practical considerations

We implemented Unico in R. In order to stabilize the parameter estimation, in practice, we consider non-negativity constraints when estimating the means and a small $L_2$ penalty when estimating the variances and covariances in the model. The latter alleviates the risk of multicollinearity and, therefore, inaccurate estimation owing to the high correlation between the proportions of different cell types. Additionally, when estimating the parameters of a given feature, we disregard samples with values that diverge from the mean by more than two standard deviations. This measure prevents extreme and non-representative data points from dominating the solution.

We optimize the Unico model iteratively. At the end of each iteration, we update the weights, which can then be used to weight the samples in the subsequent iteration (Additional file 2: Supplementary Notes). We learn the means using the constrained least squares solver `pcls` from the `mgcv` R package, and we learn the variances and covariances using the COBYLA algorithm [50] as implemented in the `nloptr` R package [51]. Empirically, we found that Unico works well using as few as two iterations (i.e., updating the weights once) for estimating the means and using three iterations for estimating the variances and covariances (data not shown).

Lastly, the Unico deconvolution estimator can occasionally yield extreme values—either positive or negative—which is an inherent property of the framework, as the non-negativity constraints imposed on the model parameters do not necessarily translate to non-negativity of the inferred cell-type level profiles. In particular, extreme negative estimates often arise from outlier input values. Because the estimator relies on bulk-level data, outliers in the bulk measurements can propagate and lead to extreme deconvolution outputs. Similarly, since the estimated effect sizes represent averages across individuals, applying them to extreme covariate values can amplify the resulting estimates. As extreme values may influence downstream analyses, we recommend handling such values using standard practices, consistent with how one would treat outlier data points in other analytical settings.

### PBMC and lung scRNA-seq data

We obtained the PBMC scRNA-seq dataset from a COVID-19 study by Stephenson et al. [30]. We arbitrarily selected only one sample for donors with multiple measurements, which resulted in a total of 118 samples for the analysis. After excluding cells with a high percentage of hemoglobin ($\geq 1\%$) or mitochondria ($\geq 5\%$), and low percentage of ribosomal content ($\leq 1\%$), in addition to requiring a minimal and maximal number of unique expressed genes ($\geq 500, \leq 2500$) and total unique molecular identifier (UMI) counts ($\geq 2000, \leq 15000$), 499,336 cells remained for the analysis. In addition, we used scRNA-seq from the data collection presented by Sikkema et al. [38] as part of a study for integrating multiple datasets collected from the human respiratory system. We focused on the lung parenchyma samples ($n = 90$) that composed most of the carefully annotated group of samples in the original study (defined by the authors as the "core reference" group). Employing the same data filtering criteria as for the PBMC data resulted in a total of 296,227 cells for the analysis. For both the PBMC and lung datasets we used the cell-type annotations provided by the authors and applied a counts per million (CPM) normalization.

### Gene expression data with follicular lymphoma

We used a preprocessed microarray bulk FL data ($n = 302$) by Newman et al. [8]. In total, out of the 302 samples available, 14 were confirmed to have the *CREBBP* mutation, and 10 samples were confirmed to exhibit a wild-type allele. The *CREBBP* status for 12 of these samples was collected by Green et al. [42] and the remaining 12 samples by Newman et al. [8]; the *CREBBP* status of all 24 samples was made available in the supplementary files of Newman et al. For defining a ground truth list of differentially expressed genes with *CREBBP* mutation in FL B cells, we considered the set of 334 up-regulated and 279 down-regulated genes that were previously reported in a study with sorted B cells from FL tumors [42]. Intersecting these sets with the genes available in the bulk FL data left us with 275 and 219 up- and down-regulated genes for evaluation. In addition, we also collected scRNA-seq dataset from tumor and immune cell populations of FL by Han et al. [43]. We excluded healthy controls and retained 122,702 cells from ($n = 20$) patients with untreated or relapsed FLs.

### Whole-blood DNA methylation datasets

We used a total of five beta-normalized DNA methylation datasets that were collected using the Illumina 450 K methylation array. For the methylation deconvolution analysis, we obtained data from Reinius et al. [39], including whole-blood ($n = 6$) and matching cell-sorted methylation data from the same individuals (granulocytes, monocytes, NK, B, CD4 T, and CD8 T cells). For the cell-type level DM analysis, we considered whole-blood datasets from Liu et al. ($n = 687$) [44], Hannum et al. ($n = 590$; samples with missing smoking status were excluded) [40], and two datasets from Hannon et al. ($n = 675$, $n = 665$) [45]. In all datasets, we removed CpGs with non-autosomal, polymorphic, and cross-reactive probes [52], and we excluded low variance CpGs (variance< 0.001). This left us with 153,155, 144,743, 134,250, and 95,360 CpGs for the Liu, Hannum, and the two Hannon datasets, respectively. For the Reinius dataset, we considered CpGs at the

intersection between the Reinius data and a preprocessed version of the Hannum dataset (restricted to samples with European ancestry; 93,086 CpGs). Lastly, cell-type proportions were estimated for all whole-blood datasets using EpiDISH, a reference-based methylation decomposition method [53].

**Implementation and application of competing deconvolution and cell-type association methods**

We ran all CIBERSORTx [8] related codes under a docker container version 1.0 encapsulating both the "High Resolution" mode (for estimating cell-type level profiles) and the "Fractions" mode (for estimating cell-type proportions) with default parameters and authentication token granted by the CIBERSORTx team upon request. CIBERSORTx evaluates the maximum value in a bulk input and automatically assumes the data have been log-normalized if the maximum is less than 50. This choice is reasonable for transcriptomic data, for which CIBERSORTx was designed; however, it is not justified for beta-normalized methylation levels that are restricted to the interval [0, 1]. We thus scaled the methylation beta values by a factor of 10,000 prior to the application of CIBERSORTx and rescaled the results back to original scale.

We installed the TCA [7] R package version v1.2.1 deposited on CRAN and evaluated its performance under default parameters. We fitted the model using the function `tca` and performed deconvolution using the `tensor` function. For the cell-type level DM analysis, both the joint (tissue-level) and marginal (cell-type level) statistical tests were automatically evaluated as part of the model parameter learning step in the `tca` function.

bMIND [10] is available via the MIND R package version 0.3.3 from CRAN. We obtained the cell-type specific profiles and the estimated model parameters with the function `bMIND` and performed association testing with the function `test`. bMIND evaluates the maximum value in the bulk input and automatically log transforms the data if the maximum is larger than 50. We, therefore, conducted two main versions of bMIND for all analyses on RNA expression datasets: one under the recommended setting where we provided both the single-cell derived prior and the bulk expression profile in log-transformed scale and another version where we did not provide any prior and kept the deconvolution in the original scale. In more detail, we achieved the latter by scaling the bulk expression profile by the inverse of the largest detected value, and then rescaling the output back to the original data scale. This approach ensured consistency and comparability across all deconvolution methods. Specifically, allowing the default log transformation of the data would have violated the assumption that bulk levels represent linear combinations of cell-type levels.

We installed the BayesPrism [12] R package v2.2.2 from CRAN. Originally designed for transcriptomics data, BayesPrism models count data and assumes $X_{ij} = \sum_{h=1}^{k} Z_{ijh}$ (considering the Unico notations). Thus, $Z_{ijh}$ in BayesPrism is in place of $W_{ih}Z_{ijh}$ in Unico. Effectively, this means that less abundant cell types will tend to have lower estimated cell-type levels. To compare BayesPrism on equal ground with the other methods, we decoupled the cell-type-level profiles and proportions by scaling each sample's estimated cell-type-level profiles across all genes. This ensured that they sum up to the total counts of the corresponding bulk profiles (i.e., $\frac{Z_{ijh}}{\sum_{j'=1}^{m} Z_{ij'h}} \cdot \sum_{j'=1}^{m} X_{ij'}$).

For analyses where we have scRNA-seq data available as prior, we used the cell state/ subtype annotation provided by the original authors. We set the cell type labels to match the granularity used for the other deconvolution methods in each analysis. Since Bayes-Prism performs the cell-type proportions and 3D tensor estimations together, we additionally provided a set of strong cell-type marker genes to improve the joint inference. Specifically, we provided BayesPrism with an additional set of the top 25 marker genes for each cell state. When deconvolving methylation levels, we used the sorted cell-type levels from Reinius et al. as prior and set the cell state and cell-type levels to be the same. To better mimic expression count data, we scaled the methylation fractions by an arbitrary large number (e.g., one million) before providing them to BayesPrism, and we added a final extra step of rescaling back to the original fraction scale.

Throughout this work, we also evaluated a baseline approach in our analysis and evaluation by simply considering the product of the observed bulk data and the cell-type proportions as cell-type level estimates. That is, we estimated $z_{ijh}$, the cell-type level of sample $i$, gene $j$, and cell type $h$ as $z_{ijh}^{\text{Baseline}} = x_{ij} \cdot w_{ih}$. Finally, we applied CellDMC [47] for DM using the implementation in the Bioconductor R package EpiDISH, version 2.10.0.

### Deconvolving mixtures of gene expression profiles and estimating cell-type level moments

We used both the PBMC and lung scRNA-seq datasets for generating pseudo-bulk mixtures. Briefly, for creating a new pseudo-bulk sample, we first drew (with replacement) all cell-type level profiles of one randomly selected sample. The cell-type profiles of each individual sample were defined as normalized pseudo-bulk counts at the cell-type level. We then drew (with replacement) the cell-type proportions of one randomly selected sample in the data (total number of cells coming from each cell type, normalized to sum up to 1). Eventually, these were used as the weights for a weighted linear combination of the cell-type level profiles to create one pseudo-bulk sample.

In the PBMC analysis, we considered either five major cell-type groups (monocytes, NK, B, CD4 T, and CD8 T cells) or seven cell types by further stratifying B cells into canonical B cells and plasma cells and monocytes into CD16 and CD14 monocytes. In the analysis with lung cells, we considered either four major cell-type groups (endothelial, stromal, immune, and epithelial cells) or six cell types by further stratifying immune cells into myeloid and lymphoid compartments and epithelial cells into the airway and alveolar epithelium cells. Our evaluation focused on the top 10,000 most highly expressed genes in the data. See Additional file 2: Supplementary Notes for more details.

To learn the underlying 3D tensors, we provided the pseudo-bulk mixtures and their corresponding mixing proportions as the input for all deconvolution methods, except for BayesPrism, which estimates mixing proportions based on the prior data. We assessed these tensors for their accuracy by comparing them against the known cell-type profiles. Particularly, for a given cell type and a given gene, we evaluated the correlation between the true cell-type expression levels of the pseudo-bulk samples and their deconvolution-based estimates.

We obtained estimates of population-level cell-type moments from the data (means, variances, and covariances per gene) directly from the output of the deconvolution methods. For methods that do not explicitly output such estimates (e.g., no method

except for bMIND and Unico outputs covariance estimates), we used the estimated tensor for calculating these moments. To evaluate the accuracy of the estimated moments, we established gold standard estimates based on the cell-type profiles underlying the pseudo-bulk mixtures. In order to mitigate the potential influence of outliers, we considered only samples within 2 standard deviations from the mean for the estimation of moments of a given gene.

Finally, we used multiple linear regression to evaluate whether an estimated 3D tensor of a given deconvolution method captures the variation of the true tensor beyond its correlation with the deconvolution input (i.e., pseudo-bulk and cell-type proportions). In more detail, for every gene and cell type, we fitted a regression model for the known cell-type expression levels as the dependent variable using several independent variables, including the pseudo-bulk levels of the gene, the cell-type proportions, and the cell-type tensor estimates. This allowed us to quantify to what extent the deconvolution-based estimates provide information beyond the bulk data. Specifically, we defined $\Delta \log_{10}(p\text{-value})$ as the difference between the log-scaled (basis 10) t-test derived $p$-values of the pseudo-bulk variable and the estimated cell-type levels in the regression. Of note, we defined the $p$-values to be 1 in cases where cell-type levels were estimated to have no variation. In order to mitigate potential biases due to heavy-tailed distributions of expression levels, we log1p-transformed expression levels and considered only samples within 2 standard deviations from the mean.

### Deconvolving the Reinius whole-blood DNA methylation data

Unlike our deconvolution of gene expression mixtures, the size of the Reinis data ($n = 6$) does not allow for drawing reliable conclusions through a straightforward evaluation. In particular, Unico and other deconvolution methods are designed to operate on large bulk data. We circumvented this limitation by taking a two-step reference-based procedure. First, we learned the parameters of the Unico model from the larger Hannum whole-blood methylation data [40]. Acknowledging that population structure affects methylation [54], we focused solely on Caucasian individuals from the Hannum data ($n = 426$), anticipating that they would adequately represent the Swedish individuals in the Reinius study. Then, we plugged these parameter estimates into Unico's 3D tensor estimator together with the Reinius bulk profiles and their cell-type proportion estimates. We performed the same procedure for TCA, however, CIBERSORTx and bMIND, which do not provide an analytical estimator of the tensor, required a different strategy. In order to inform the deconvolution of CIBERSORTx and bMIND with external information, we applied these methods to the concatenation of the Reinius and Hannum datasets and extracted the cell-type level estimates for the Reinius samples.

Due to bMIND's Bayesian framework, we also explored another approach, denoted as "bMIND (prior)", in which we used parameter estimates from the Hannum data as a prior to inform its parameter learning and deconvolution of the Reinius bulk profiles. BayesPrism's framework requires us to use explicit cell-type level data as prior. We set the cell-type profiles from the Reinius data as prior. Since we also used these profiles as the ground truth for evaluation, we refer to this strategy as "BayesPrism (oracle prior)", indicating it represents the best-case scenario for BayesPrism. In that case, we added CpGs previously identified as informative for capturing cell-type heterogeneity [55] to

aid BayesPrism's joint estimating of the tensor and the cell-type proportions. For completeness, we also applied a similar treatment to bMIND and Unico, by setting the prior of the former and the model parameters of the latter based on the estimates from the cell-type level ground truth data. These are referred to as "bMIND (oracle prior)" and "Unico (oracle prior)", respectively (Additional file 1: Figs. S21 and S23).

Benchmarking methods based on the Reinius data presents a second challenge: determining a proper way to evaluate their performance given that data from only six individuals is available for the analysis. We tackle this limitation by collapsing methylation levels in the estimated tensor along both the CpG and sample axes. That is, for every cell type, we evaluated how correlated is the vector of all methylation estimates of the cell type (i.e., by pooling estimates across all CpGs and samples) with the experimentally measured ground truth levels from purified cells. This yielded a single correlation score per cell type. Importantly, when stacking CpGs for evaluation, a deconvolution that only correctly estimates relative means and scales of CpGs but performs poorly in terms of per-CpG correlation (i.e., across samples) may achieve spuriously high correlation levels. We addressed this by removing from every CpG its cell-type-level mean methylation.

Since beta-normalized methylation levels are bounded to the range [0,1], unlike in the deconvolution of relative expression levels, we further evaluated the divergence of the estimated 3D tensors from the true cell-type levels in absolute terms. Specifically, we evaluated the root median square error (RMSE) between the true and each estimated 3D tensor; we expected that a median metric in place of a standard mean square error would improve robustness to outliers. Similarly to the evaluation of correlation, we calculated an RMSE value per cell type after collapsing methylation levels in the tensors along both the CpGs and samples axes.

Finally, our benchmarking focused either on a set of randomly selected CpGs or on a set of highly variable CpGs based on the Reinius data. For defining the latter, we ranked the CpGs, in the intersection of the Reinius and Hannum datasets (93,086 CpGs), by the sum of their variances in different cell types calculated using the sorted Reinius dataset and chose the top 10,000 CpGs with the largest values.

### Calculating robust linear correlation

All the correlation values reported throughout our analysis and evaluation were calculated using a robust linear correlation metric in place of the standard Pearson correlation. Specifically, we used the function `cov.rob` from the MASS R package [56], which performs an approximate search for a subset of the observations to exclude, such that a Gaussian confidence ellipsoid is minimized in volume. Effectively, this procedure trims outliers that may otherwise dramatically bias correlation levels (Additional file 1: Fig. S33). In particular, if either input vector has an interquartile range (IQR) of 0, `cov.rob` defines the correlation as 0. Throughout the paper, we set the outlier exclusion threshold to 5% of the data points.

### Calculating von Neumann entropy

We quantify the amount of signal coming from the covariance structure of a given gene by the von Neumann entropy [31]. For a given gene, the von Neumann entropy is defined as the entropy applied to the eigenvalues of the normalized cell-type covariance

matrix of the gene (i.e., a $k \times k$ matrix of correlations between cell types). High entropy corresponds to cases where no substantial cell-type covariance structure exists, and low entropy indicates strong positive or negative correlations between cell types. Throughout our evaluation of the deconvolution results, we grouped genes into high- and low-entropy groups. This classification was based on ranking the genes by their entropy and assigning genes with above-median (below-median) entropy to the high- (low-) entropy group. Lastly, the normalized von Neumann entropy presented in Fig. 1b simply refers to von Neumann entropy values scaled to the range [0,1]. Since the von Neumann entropy is bounded by a number that depends on the number of cell types $k$, this normalization enables us to evaluate and visualize the distribution of entropy across genes using covariance matrices of different sizes.

### Deconvolving bulk profiles from follicular lymphoma tumors

For every deconvolution method, we first estimated the 3D tensor of the bulk FL dataset ($n = 302$) while considering only the sets of 275 and 219 genes that were previously reported as up- and down-regulated with the *CREBBP* mutation. We provided each method with cell-type proportions estimated using CIBERSORTx ("Fractions" mode) with the LM22 signature matrix [57], while collapsing the estimated proportions into 4 categories: B cells, CD4 T cells, CD8 T cells, and "remaining."

A straightforward evaluation would include calculating for every method log-fold changes (LFCs) with the *CREBBP* mutation based on the estimated B cell expression levels. This would allow for assessing the concordance between the LFCs and the previously reported direction of the differentially expressed genes. However, the group of *CREBBP*-mutated tumors presents an elevated B cell composition, which is expected to lead to an overly optimistic performance on the set of up-regulated genes in cases of deconvolution estimates that are biased by cell-type composition (Additional file 1: Fig. S34). Most notably, since the baseline method estimates B cell expression levels by naively multiplying bulk levels and B cell proportions, the baseline estimates are expected to be artificially higher for samples with higher B cell composition. The baseline method, therefore, consistently estimates higher B cell expression levels for the *CREBBP*-mutated tumors, regardless of whether the genes are truly down- or up-regulated. Consequently, genes that are truly up-regulated in *CREBBP* tumors are expected to present strong LFCs under the baseline given the combination of both real and artificial up-regulation effects.

In order to account for the B cell composition bias, we used multiple linear regression to test whether the estimated tensors capture the mutation effects beyond the effect of B cell composition. In more detail, for every gene, we fitted a regression model for the estimated B cell expression levels as the dependent variable using the B cell composition and the mutation status as independent variables. We performed the same procedure while using the bulk expression levels as the dependent variable to evaluate a standard analysis of bulk expression. To facilitate a comparable evaluation of the estimated mutation effect sizes across different methods and to mitigate the potential impact of outliers, we standardized the log1p-scaled B cell expression estimates for each gene. For methods that do not constrain non-negativity in their estimated tensor, for every gene and cell type, we shifted the distribution of the estimates by subtracting the minimum value detected, which enforced non-negativity prior to the log1p transformation. The effect

sizes of genes that were estimated to have a constant B cell expression level across all samples were set to 0.

### Cell-type level epigenome-wide association studies with sex and age

We performed statistical testing for calling DM using TCA [7], CellDMC [47], Unico (distribution-free), and a parametric version of Unico (Additional file 2: Supplementary Notes). For bMIND [10] and BayesPrism [12], we provided cell-type level methylation profiles from Reinius et al. as prior (Additional file 2: Supplementary Notes). As a baseline model, we evaluated the linear effects of the conditions on the tensor estimates of our baseline deconvolution. Concretely, for a given CpG and cell type, we fitted a linear regression model with the baseline-estimated cell-type level methylation as the dependent variable and the condition (and covariates) as the independent variable. This allowed us to calculate t-statistics and derive *p*-values for the cell-type level effects of the conditions under a baseline deconvolution.

Our analysis included cell-type- and tissue-level covariates ($\{c_i^{(1)}, c_i^{(2)}\}$; see Additional file 2: Supplementary Notes). For cell-type covariates, we considered age and sex in the analysis of all four whole-blood methylation datasets (Liu et al. [44], Hannum et al. [40], and two cohorts by Hannon et al. [45]). In addition, we accounted for rheumatoid arthritis and smoking status in the Liu data, schizophrenia status in the Hannon data, and ethnicity and smoking status in the Hannum data. Across all datasets, smoking status was classified into three major categories: never, past, and current smoker. For tissue-level covariates, we considered surrogates of technical variability. In more detail, for each methylation dataset, prior to filtering any CpG, we took a previously suggested approach [7, 29] of estimating factors of technical variation by calculating the top 20 principal components (PCs) of the 10,000 least variable CpGs of each methylation array. We expected these PCs to capture only global technical variation and no biological variation due to the use of CpGs with nearly constant variance. In addition to these PCs, we further accounted for plate information, which was available for the Hannum data. All the benchmarked methods were designed to account for cell-type and tissue-level covariates, except for the baseline model. For the latter, we simply included the full set of covariates as independent variables in the linear regression model.

The inter-individual distribution of array-probed methylation levels is approximately normally distributed for most CpGs. For that reason, TCA and CellDMC, which were designed for methylation data, assume the data is normally distributed; bMIND assumes normality as well, even though it was not designed for methylation.

For any given ordered pair of datasets (discovery and validation), we considered the CpGs at the intersection of the two datasets. True positives (TPs) were defined as CpGs that are (i) genome-wide significant in the discovery dataset under a Bonferroni-corrected threshold and (ii) significant in the validation dataset, under a Bonferroni-corrected threshold adjusting for the number of significant hits identified in the discovery data. CpGs that only satisfied condition (i) and either failed to satisfy condition (ii) or demonstrated inconsistent direction of their estimated effect size were considered false negatives (FNs). CpGs with *p*-value> 0.95 in the discovery dataset were considered as negative controls for the evaluation of false positives (FPs) and true negatives (TNs). That is, negative controls with

significant (non-significant) *p*-values under a Bonferroni-corrected threshold adjusting for the number of negative controls in the validation data were counted as FPs (TNs).

Finally, as a metric of consistency across datasets, we calculated the MCC per method for every pair of discovery and validation datasets. We favored MCC over the widely-used *F*1 score since the former incorporates true negatives, which makes it a better choice for assessing model performance on imbalanced class distributions [46]. Yet, for completeness, we further considered the *F*1 score as the consistency metric, which revealed qualitatively similar results (Additional file 1: Figs. S35–S38).

## Supplementary Information

---

Additional file 1. Supplementary Figs. S1-S38.

Additional file 2. Supplementary Notes.

Additional file 3. Peer review history.

---

**Acknowledgements**
Not applicable.

**Review history**
The review history is available as Additional file 3.

**Peer review information**

**Authors' contributions**
ZJC and ER conceived and developed the Unico model. ZJC implemented the model, benchmarked it, and generated the figures. ER and EH provided input and feedback and supervised the study. All authors wrote, read, and approved the final manuscript.

**Funding**
ZJC, ER, and EH were supported by the National Human Genome Research Institute grant R01HG010505.
ER was also supported by the National Human Genome Research Institute grant 1R21HG013393. EH was also supported by the National Institute of Biomedical and Bioengineering grant R01EB035028.

**Data availability**
All datasets supporting the conclusions of this article are publicly available. The bulk FL data is available from Gene Expression Omnibus (GEO) accession number GSE127462 [8, 58]. The whole-blood methylation data with matching sorted cells, as well as the whole-blood methylation datasets used for cell-type level DM analysis are available from GEO (accessions GSE35069 [39, 59], GSE40279 [40, 60], GSE42861 [44, 61], and GSE80417 [45, 62], GSE84727 [45, 62]). The PBMC scRNA-seq dataset [30] was downloaded from European Molecular Biology Laboratory's European Bioinformatics Institute (EMBL-EBI) accession E-MTAB-10026 [63]. The lung scRNA-seq dataset [38] is available on CELLxGENE [64] as "The integrated Human Lung Cell Atlas" [65]. The FL tumor scRNA-seq dataset [43] is available on CELLxGENE [64] as the "Single cell RNA sequencing of follicular lymphoma" [66]. Unico is available under GNU General Public License 3.0 as an R package on CRAN, at GitHub https://github.com/cozygene/Unico [67], as well as Zenodo https://doi.org/10.5281/zenodo.13952730 [68].

## Declarations

**Ethics approval and consent to participate**
Not applicable.

**Consent for publication**
Not applicable.

**Competing interests**
The authors declare that they have no competing interests.

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

## 