## [Additional file 3. Peer review history. · Genome Biology]

Review history

First round of review

Reviewer 1

Chen et al. used the method of moments to deconvolve bulk gene expression or DNA methylation data to estimate sample-level cell type-specific (CTS) omics. They claimed the proposed Unico method is distribution-free, but used normal distribution assumptions in CTS-DM. The method and algorithm are rigorously developed. They compared Unico with several existing methods via simulations using single-cell PBMC and lung datasets. They calculated the effect size for CTS-DE and tested CTS-DM. The paper is well written. I provide some major comments below that must be addressed.

1. Fig 2b (deconvolution of bulk gene expression): The simulations are not very thorough, for instance they do not include BayesPrism, which is a highly cited paper
<https://www.nature.com/articles/s43018-022-00356-3>

2. Of greater concern is that the simulation results contradict the literature, e.g., bMIND has repeatedly been verified by multiple groups to work better than TCA in brain and heart tissues. In the EPIC-unmix algorithm, bMIND has been improved in blood with the updated priors learned from the initial estimates.

<https://www.science.org/doi/10.1126/sciadv.adh2588>

<https://genome.cshlp.org/content/31/10/1807.short>

<https://doi.org/10.1093/bib/bbad273>

<https://www.biorxiv.org/content/10.1101/2024.05.23.595514v1>

To resolve the discrepancy with the literature, the authors must explore more tissue types/datasets and make the simulation more realistic by directly combining single-cell counts to generate pseudo-bulk data without other arbitrary treatments.

3. It is also essential to test existing methods under the recommended settings where they were designed. There are two ways in which bMIND was not used as would be expected.

3a. First, other than rescaling the RNA-seq data with the largest value, the authors should run bMIND with log-transformed data. The authors say, "Specifically, allowing the default log transformation of the data would have violated the assumption that bulk levels represent linear combinations of cell-type levels." Regardless, this is how the method was designed and it is not up to the authors to change the implementation.

Let us revisit the Zhong and Liu's criticism of the utilization of variance stabilizing transformations, such as log-scaling (Ref 33). They say this would violate the linearity assumption for convolution. Of course this is true, however, when they made this claim, which was targeted at Shen-Orr, S.S. et al. Nat. Methods 7, 287-289 (2010), Shen-Orr et al. rebutted the complaint. In essence, they argue that, yes the assumption of linearity after transformation appears to be false; however, in practice the linearity assumption works well, much better than alternatives that attempt to deconvolve in the original scale. Their point was that a little bias is worthwhile, when it greatly reduces the variance of the estimator.

3b. Second, one almost always has prior data from the web. While bMIND runs without a single cell

prior, it is unlikely that anyone would chose this option giving the easy availability of relevant data sources. The comparisons should feature the most common usage of competing methods.

4. Fig 2de (deconvolution of bulk DNA methylation) is based on six samples. To make the simulation more reliable, as with resampling scRNA-seq data, the authors can resample publicly available single-cell DNA methylation (DNAm) to generate a much larger sample, or construct bulk samples with larger sorted-cell DNAm datasets.

To calculate the CTS DNAm for six samples, the authors estimated the model parameters for TCA and Unico and drew 3D tensors from the model with bulk data. This is equivalent to using a strong prior with infinite confidence.

However, "in order to inform the deconvolution of CIBERSORTx and bMIND with the same additional information, we applied these methods to the concatenation of the Reinius and Hannum datasets and extracted the cell-type level estimates for the Reinius samples."

The simple concatenation of the Reinius (n = 6, to be deconvolved) and Hannum (bulk data where TCA and Unico learned parameters) datasets is not an efficient way to use the info from Hannum data. At least bMIND can take the learned mean and covariance parameters as priors.

5. Fig 3a (CTS-DE), it makes sense to only adjust for B cell composition for bulk baseline, NOT for other deconvolved expressions, since the cell compositions have been used and accounted for in the deconvolution.

6. Fig 3b (CTS-DM), "We, therefore, similarly applied statistical testing under a normality assumption when evaluating Unico on calling DM. Notably, this assumption is not required given that the Unico framework is generally distribution-free and allows us to derive asymptotic p-values."

Using the normal assumption for testing contradicts Unico's distribution-free claim. Why not directly use Unico's asymptotically derived p-values under non-parametric testing if the parametric normal assumption is not required? In addition to CTS-DM, the properties of Unico's testing for CTS-DE should be assessed, other than comparing effect size in Fig 3a.

7. If the model direction of testing really matters ($Y|X$ or $X|Y$, Ref 28), the proposed testing procedures of Unico seem not suitable to identify causal genes/CpGs that cause diseases since it assumes covariates affect methylation. If so, this is a significant limitation since identifying disease-causing features is more interesting than testing covariates.

8. The paper lost much information by restricting to 10,000 most expressed genes and 10,000 most variable CpGs.

9. The authors claim: "There are currently no large publicly available bulk datasets with matching cell-type level data for the same group of individuals."

This is not true. Bulk and scRNA-seq data from over 400 ROSMAP donors are available on Synapse.

10. Minor: Line 110, "cell-type level means and covariates", "covariates" should be "covariances".

Reviewer 2

This paper presents a method called Unico that is meant to estimate the pure cell type expression matrix given only the bulk expression and cell-type proportions. The authors demonstrate that several of the parameters estimated by Unico are more accurate than competing methods.

The analysis uses a suboptimal "baseline" which the authors define as "naively weighting each bulk profile by the cell-type proportions of the sample" This baseline is uniformly worse than other baselines namely: raw bulk expression and bulk expression with cell type proportions regressed out. These baselines must be included for a proper analysis.

When we run this analysis using the supplied tutorial code, which we can assume represents the best scenario performance for this method it can be shown that the estimates produced are very similar across cell types in terms of inter-subject variation and are in fact no better than a proportion corrected to the bulk gene expression. This analysis is detailed in the attached PDF.

The statement "This advancement enhances our capability to conduct powerful large-scale genomic studies at cell-type resolution without the need for cell sorting or single-cell biology." is not supported by the results. The deconvolution problem is clearly not identifiable when using cell proportions and even if reference is included, as is the case with other methods. It is still possible that non-trivial solutions can be obtained in some cases. It would encourage the authors to be more explicit about how and why their method achieves this goal.

Minor comments

- the benchmarking results from page 8 is using ground truth fraction as input, which in real case you don't have access to. Meanwhile, for 'cibersortx' method it not clear what fraction it is using
- in page 22 last sentence, they indeed applied non-negativity constraint, but it is for mean only, the deconvolved Z can still contain negative values
- I don't see much difference between methods in Fig.3a
- page 31, using cbsx LM22 as signature for fraction deconvolution results can be unreliable, since LM22 is for normal PBMC; if possible should use other targeted reference

Authors' response to reviewers

We thank the reviewers for their comments and suggestions. We have made a considerable effort to address the reviewers' comments by running additional analysis and adjusting our manuscript.

Specifically, we:

1. Added BayesPrism to all of our benchmarkings.
2. Considered multiple versions of bMIND, including one with single-cell prior information while using default settings (in particular, we used log-transformed expression data as input).
3. Highlighted the performance of Unico under its distribution-free assumption in the cell-type level differential methylation study.
4. Revised the text in the manuscripts to clarify and address the reviewers' questions and suggestions.

Below, we provide a point-by-point response to the reviewers' comments (our responses are in blue font).

Reviewer #1: Chen et al. used the method of moments to deconvolve bulk gene expression or DNA methylation data to estimate sample-level cell type-specific (CTS) omics. They claimed the proposed Unico method is distribution-free, but used normal distribution assumptions in CTS-DM. The method and algorithm are rigorously developed. They compared Unico with several existing methods via simulations using single-cell PBMC and lung datasets. They calculated the effect size for CTS-DE and tested CTS-DM. The paper is well written. I provide some major comments below that must be addressed.

1. Fig 2b (deconvolution of bulk gene expression): The simulations are not very thorough, for instance they do not include BayesPrism, which is a highly cited paper

<https://www.nature.com/articles/s43018-022-00356-3>

We thank the reviewer for pointing out that BayesPrism should also be considered in our benchmarking. We added BayesPrism, a prior-based deconvolution method, to our benchmarking. Our results show that the deconvolution estimates of Unico are more accurate than those of BayesPrism despite providing the latter with a prior. We observe this in gene expression deconvolution based on single-cell mixtures (wherein an "oracle" prior based on the same single-cell data used for simulating bulk mixtures was provided to BayesPrism; Supplementary Figures S2 and S3). Similarly, in DNA methylation deconvolution, when the experimentally measured cell-type profiles used for benchmarking were provided as prior for both Unico and BayesPrism, Unico's tensor estimates achieved better RMSE and correlation against ground truth (Methods; Supplementary Figures S15 and S17).

Unico also outperformed BayesPrism in downstream analysis. In the analysis of B cell expression with follicular lymphoma, Unico significantly outperformed BayesPrism (taking external single-cell data as prior) in recapitulating the effects identified in expression from sorted B cells (Figure 3a). Similarly, in the cell-type level DM analysis, Unico demonstrates better consistency than BayesPrism when the latter was provided with experimentally sorted cell-type profiles as prior information (Supplementary Figures S22 and S30).

2. Of greater concern is that the simulation results contradict the literature, e.g., bMIND has repeatedly been verified by multiple groups to work better than TCA in brain and heart tissues. In the EPIC-unmix algorithm, bMIND has been improved in blood with the updated priors learned from the initial estimates.

<https://www.science.org/doi/10.1126/sciadv.adh2588>

<https://genome.cshlp.org/content/31/10/1807.short>

<https://doi.org/10.1093/bib/bbad273>

<https://www.biorxiv.org/content/10.1101/2024.05.23.595514v1>

To resolve the discrepancy with the literature, the authors must explore more tissue types/datasets and make the simulation more realistic by directly combining single-cell counts to generate pseudo-bulk data without other arbitrary treatments.

In our gene expression simulations, bMIND does work better than TCA when using a single-cell prior and allowing the default log transformation for the former, as suggested by the reviewer in the next comment (however, not when applying bMIND without prior and using the original data scale). Figure R1 below shows the relative performance between these methods, confirming that – consistent with the existing literature – bMIND indeed performs better than TCA in this task. Thus, there is no discrepancy between our results and the literature.

We note that in the paper, we first focus on methods that do not leverage prior information (Figure 2a-c), and only then describe the performance of methods with prior (i.e., bMIND and BayesPrism;

Supplementary Figures S2 and S3). Therefore, bMIND with prior and TCA are not described in the same figure. We believe this structure would make it easier for the readers to follow (See “Establishing a new state-of-the-art deconvolution for bulk genomics” under the Results section).

Figure R1. Evaluation of the concordance between the deconvolution estimates and the known cell-type profiles across 20 sets of pseudo-bulk mixtures based on PBMC scRNAseq profiles of five cell types (500 samples and 600 randomly selected genes in each set). Boxplots reflect the distribution of linear correlation across all genes, and percentages indicate average cell-type abundances.

The reviewer further commented on the design of our simulation. However, any simulation provides a limited view, and any of the countless design choices and parameters can be fairly criticized.

Different choices can lead to differences in performance and evaluation between studies. For example, previous benchmarks did not consider robust correlation, which we show can be critical (Supplementary Figure 27). Similarly, the implementation/application of the different methods may also have a dramatic effect on performance. For example, in the bMIND paper and the Dai et al. paper the reviewer referred to, the authors used Bisque (Jew et al., 2020 Nature Communications) for estimating cell-type proportions as part of their benchmarking. Yet, Bisque was shown to perform poorly in some scenarios ("BayesPrism" Chu et al., 2022 Nature Cancer, Chen et al., 2022 Nature Communications, and more recently Ivich et al., 2024 bioRxiv), suggesting it might dramatically bias evaluations that rely on Bisque proportions.

Despite their limitations, we believe simulations may suggest insight into the dynamics of the different methods as we vary the parameters of the simulations. However, for the reasons above, the conclusions in our paper mostly rely on the results of analyzing multiple real datasets.

3. It is also essential to test existing methods under the recommended settings where they were designed. There are two ways in which bMIND was not used as would be expected.

3a. First, other than rescaling the RNA-seq data with the largest value, the authors should run bMIND with log-transformed data. The authors say, "Specifically, allowing the default log transformation of the data would have violated the assumption that bulk levels represent linear combinations of cell-type levels." Regardless, this is how the method was designed and it is not up to the authors to change the implementation.

Let us revisit the Zhong and Liu's criticism of the utilization of variance stabilizing transformations, such as log-scaling (Ref 33). They say this would violate the linearity assumption for convolution. Of course this is true, however, when they made this claim, which was targeted at Shen-Orr, S.S. et al. Nat. Methods 7, 287-289 (2010), Shen-Orr et al. rebutted the complaint. In essence, they argue that, yes the assumption of linearity after transformation appears to be false; however, in practice the linearity assumption works well, much better than alternatives that attempt to deconvolve in the original scale. Their point was that a little bias is worthwhile, when it greatly reduces the variance of the estimator.

3b. Second, one almost always has prior data from the web. While bMIND runs without a single cell prior, it is unlikely that anyone would chose this option giving the easy availability of relevant data sources. The comparisons should feature the most common usage of competing methods.

Under our simulation framework, a violation of the linear assumption drastically skews bMIND's estimates of means, variances, and covariances. In our original submission, we felt that keeping the data at the "right scale" (with respect to the simulated data) is more reflective of bMIND's performance (which otherwise looks poor, at least based on the estimates of means, variances,

and covariances; Supplementary Figures S2 and S3). However, we agree with the reviewer that the primary evaluation of a method should be under the default/author-recommended settings. Also, we agree that violating a theoretical assumption may not necessarily correspond to worse performance empirically.

We now include in all of our experiments the results of applying bMIND using prior based on single-cell data and log-scaling the gene expression levels (i.e., recommended settings). Specifically, in the simulation of bulk mixtures, we provide bMIND with the ground truth prior ("oracle prior", based on the same data used for creating the bulk mixtures), and in the B cell analysis, we provide bMIND with single-cell data from Han et al., 2022 Blood Cancer Discov. In addition, as before, we provide a version of bMIND without prior and without log-transforming the data, which reflects a version of bMIND that is more on par with Unico. We believe it is important to include the latter, especially since in some cases, the data used as prior may be of poor quality or may not represent well the study population. Avoiding a log-transformation further allows bMIND to estimate moments that are calibrated with the simulation's moments (Figure 2a). In the deconvolution of DNA methylation, we evaluate bMIND using different priors, including an "oracle prior" based on sorted cells.

In all cases, Unico performs better (or as well) as bMIND with prior, including in the deconvolution and downstream analysis of gene expression (Figure 3a; Supplementary Figures S2 and S3) and the deconvolution and downstream analysis of DNA methylation (Supplementary Figures S15, S17, S22, and S30). We have updated the text throughout the manuscript accordingly.

Finally, we agree with the reviewer that applying variance-stabilizing transformations may empirically perform better than using the original scale of the data, especially in the context of gene expression data. Our original intention was to motivate the need for models that can account for the right distribution of the original scale of the data (rather than assuming, for example, that the original scale is normal). Such models could, in principle, avoid biases introduced by data transformations. In response to the reviewer's comment, we modified our text, which now reads:

"Current deconvolution methods can be categorized into two groups: heuristic approaches, including CIBERSORTx and CODEFACS, and parametric methods, which assume the observed data follows a predefined distribution, including TCA, MIND, bMIND, and BayesPrism. Parametric methods are susceptible to deviations from their underlying distributional assumptions. For instance, the common assumption that data follows a normal distribution is invalid for transcriptomic data. While

variance-stabilizing transformations like log-scaling can empirically mitigate this issue in some cases, they also break the linearity assumptions in Equations 1-2, leading to biased estimates. A more principled approach that accounts for the actual distribution of the data may yield more accurate and reliable results."

4. Fig 2de (deconvolution of bulk DNA methylation) is based on six samples. To make the simulation more reliable, as with resampling scRNA-seq data, the authors can resample publicly available single-cell DNA methylation (DNAm) to generate a much larger sample, or construct bulk samples with larger sorted-cell DNAm datasets.

First, we would like to emphasize that our results in Figure 2d,e are not based on simulations. The Reinis et al. data used in this analysis includes whole-blood methylation and matching,

experimentally-sorted cell-type methylation from several cell types. The experimentally validated cell-type profiles in this dataset provide a unique and unbiased opportunity for evaluation. This dataset is indeed limited in the number of samples (n=6). We addressed this limitation by evaluating the different methods not only for the accuracy of their deconvolution estimates across samples but also across CpGs (i.e., deconvolution estimates of a specific cell type across several CpGs), which allowed us to establish robustness and statistical significance.

Unfortunately, we could not simulate reliable methylation bulk mixtures similar to our gene expression simulation. A key message of our manuscript and a key element of Unico is the additional information we can obtain by modeling cell-type covariances (also modeled in bMIND, as we indicate in our manuscript). Mixing cell-type-level profiles not measured in the same sample (individual) would break the real covariance structure between cell types. Our simulations are, therefore, limited in their “sample dimension” to the same number of samples in the dataset with cell-type-level profiles. Put differently, the “effective” number of samples in the simulated mixtures will still be very small if we only have a small number of samples with cell-type profiles.

Existing single-cell methylation datasets include only a handful of samples, and while several large datasets with methylation from sorted cells exist, these datasets purified only one or two cell types (e.g., Reynolds et al. 2014 Nature Comm). To the best of our knowledge, the Reinis et al. data with experimentally sorted cell-type methylation from several cell types (and matching whole-blood methylation) is the most suitable data for evaluating the different methods without breaking the real cell-type methylation covariance structure. We could, in principle, simulate data based on the cell-type-level profiles in the Reinis data. However, simulating larger datasets based on this data would still have an effective sample size of six. In contrast, our gene expression simulations are based on a much larger number of single-cell samples (n=118 in the PBMC data and n=90 in the lung data).

For these reasons, we provide a comprehensive evaluation of cell-type EWAS across four independent datasets and two covariates. The results of this analysis (Figure 3b and Supplementary Figures S22, S25, S29, S30, and S32) provide strong evidence that Unico significantly improves upon alternative methods in the deconvolution of DNA methylation. In the Discussion section, we further comment on why we believe downstream analysis (as opposed to directly evaluating the deconvolution estimates) is more important for benchmarking the performance and utility of deconvolution methods:

“Evaluating the correlation of deconvolution estimates with cell-type patterns merely represents an indirect (yet common) metric for comparing deconvolution methods. Such correlations rely on point estimates of the cell-type patterns, which are expected to be noisy and reflect the “best guess” of a given model. Yet, deconvolution methods such as Unico also model the uncertainty of those point estimates.

This uncertainty can be integrated to perform well-powered downstream statistical analysis without explicitly generating noisy point estimates. As we demonstrate through the analysis of multiple RNA and DNA methylation datasets, this approach advances over a standard bulk analysis in terms of standard performance metrics and its ability to report results at a cell-type resolution.”

To calculate the CTS DNAm for six samples, the authors estimated the model parameters for

TCA and Unico and drew 3D tensors from the model with bulk data. This is equivalent to using a strong prior with infinite confidence. However, "in order to inform the deconvolution of CIBERSORTx and bMIND with the same additional information, we applied these methods to the concatenation of the Reinius and Hannum datasets and extracted the cell-type level estimates for the Reinius samples."

The simple concatenation of the Reinius (n = 6, to be deconvolved) and Hannum (bulk data where TCA and Unico learned parameters) datasets is not an efficient way to use the info from Hannum data. At least bMIND can take the learned mean and covariance parameters as priors.

The decision to concatenate the Reinius data with Hannum for evaluating CIBERSORTx and bMIND was based on our intuition that this should be at least as good as providing a prior based on Hannum (at least for bMIND, which can incorporate such a prior). However, we agree with the reviewer about the need to additionally evaluate bMIND with a prior based on Hannum (i.e., based on the learned mean and covariance parameters) instead of data concatenation.

We now show the performance of bMIND using prior based on the Hannum data (Figure 2b), as well as the performance of bMIND using prior based on the cell-type profiles in the Reinius et al. data (i.e., the ground truth cell-type profiles, or "oracle prior"; Supplementary Figures S15 and S17). Our results show that Unico outperforms bMIND in all cases. We have updated the text to reflect these additional results.

"Incorporating the ground truth cell-type profiles as prior information, reflecting the best-case scenario, allowed BayesPrism (but not bMIND) to perform better than Unico; however, incorporating the same information in Unico's deconvolution (Methods) outperformed BayesPrism's average correlation by 15.7%"

5. In Fig 3a (CTS-DE), it makes sense to adjust only for B cell composition for the bulk baseline, NOT for other deconvolved expressions, since the cell compositions have been used and accounted for in the deconvolution.

We agree with the reviewer's conceptual point. That is, we do not need to account for cell composition, given cell-type level methylation. However, we evaluate computationally deconvolved levels rather than real cell-type levels. Deconvolution methods use the cell-type composition as information for the deconvolution, which means the output deconvolution estimates are a function of the cell-type composition. Empirically, we observe that the deconvolution output of all methods indeed correlates with the cell-type composition.

In the analysis the reviewer refers to (Figure 3a), cell-type composition forms a strong confounding effect due to an increase in B cell composition with the CREBBP mutation (e.g., a simple baseline model appears to outperform all deconvolution methods in identifying upregulated genes; Supplementary Figure S28b). Thus, not accounting for residual B cell composition variation can bias the results. Specifically, a deconvolution method that artificially increases the correlation of its output with the B cell composition will perform well in this task, regardless of the correlation between its output and the true cell-type profiles.

Moreover, if the deconvolution outputs are independent of the cell composition, then accounting for the latter should not make a difference in our evaluation.

6. Fig 3b (CTS-DM), "We, therefore, similarly applied statistical testing under a normality assumption when evaluating Unico on calling DM. Notably, this assumption is not required given that the Unico

framework is generally distribution-free and allows us to derive asymptotic p-values."

Using the normal assumption for testing contradicts Unico's distribution-free claim. Why not directly use

6.1 : Unico's asymptotically derived p-values under non-parametric testing if the parametric normal assumption is not required?

We thank the reviewer for raising this important point. In the previous version of the manuscript, we compared the statistical testing (cell-type EWAS) of different methods under the same normality assumption for all methods. We supplemented this with multiple (supplementary) figures showing a near-identical performance across Unico with the parametric normality assumption and Unico with a distribution-free assumption (i.e., asymptotic p-values). We hoped this presentation would improve clarity and provide insight into the performance of Unico. However, in light of the reviewer's comment, we agree this presentation should be adjusted.

We revised the main figure to include results based on both Unico's asymptotic p-values and Unico's parametric p-value under the normality assumption (Figure 3b), and we revised the text accordingly. We now clarify that our distribution-free evaluation is the main evaluation, and the parametric evaluation is provided as an additional evaluation. We believe the latter is valuable: given that array-probed DNA methylation is known to be well-approximated by a normal distribution, one may wonder whether Unico's improvement upon the other methods in the context of methylation (which assume normality) can be primarily attributed to the other properties of Unico (i.e., modeling cell-type covariance). In addition to Figure 3b, as before, we provide supplementary figures directly demonstrating the high concordance between the parametric and distribution-free p-values of Unico (Supplementary Figures S23, S24, and S29).

6.2: In addition to CTS-DM, the properties of Unico's testing for CTS-DE should be assessed, other than comparing effect size in Fig 3a.

We tried additional evaluations of the results reported in the B cell analysis (Figure 3a). Specifically, we quantified the numbers of true positives and false negatives of each method. However, the small sample size (n=24 with CREBBP mutation status) does not allow us to provide strong statistical evidence supporting which method(s) perform the best. In our evaluation, we addressed this limitation by evaluating effect sizes across hundreds of genes, which allows us to establish statistical significance for the improvement of Unico.

7. If the model direction of testing really matters (Y|X or X|Y, Ref 28), the proposed testing procedures of Unico seem not suitable to identify causal genes/CpGs that cause diseases since it assumes covariates affect methylation. If so, this is a significant limitation since identifying disease-causing features is more interesting than testing covariates.

The reviewer refers to the two different model directionalities extensively discussed in Reference 28. Some deconvolution models, such as TCA and bMIND, provide an interface for statistical testing of X|Y models (i.e., testing a covariate/phenotype for its effect on methylation levels). However, all deconvolution methods can, in principle, be tested under the X|Y assumption, even methods that do not provide an explicit interface for that, such as CIBERSORTx. This can be achieved by using the deconvolution estimates as the explained variable in a regression model (more discussion is available in Ref 28).

TCA and bMIND also provide an interface for statistical testing under a Y|X model (i.e., a

phenotype is modeled as affected by cell-type-level methylation, which is the opposite direction of an X|Y model). While TCA allows this without explicitly learning the cell-type level profiles (i.e., the Z tensor), all deconvolution methods we evaluated, except for CellDMC (which is a decomposition method, not a deconvolution method), do allow a Y|X model: one can simply learn the tensor Z and use all the deconvolution estimates (of all cell types) of the specific CpG under test as the explaining variables in a standard regression analysis.

That said, properly testing a Y|X model without explicitly learning the cell-type level profiles can have the desired property of integrating over the uncertainty of the tensor Z, which, in theory, may perform better (see Reference 28 for an extensive discussion). Unlike in TCA, for example, the existing Unico framework does not allow us to readily derive a distribution-free Y|X model that does not require explicit estimates of Z. Such a model would require developing a completely new optimization procedure, which must be evaluated thoroughly. We, therefore, consider this to be out of scope for this paper. We added to the Discussion section text to indicate that a Y|X model is also possible with Unico, and one might gain by integrating over the uncertainty of the underlying tensor.

"Testing for differential cell-type variation within our framework models the effect of covariates and phenotypes on cell-type genomic variation without relying on point estimates of cell-type patterns. Unico can also be applied to model the opposite direction (i.e., the effect of cell-type genomic variation on a phenotype). However, unlike TCA and bMIND, this currently requires using point estimates of the cell-type patterns as the explaining variables within a regression framework. Given recent findings that highlight how properly specifying the direction of the model can significantly influence the statistical properties of hypothesis testing, future efforts could potentially enhance power and calibration by extending Unico to learn the opposite direction without relying on explicit point estimates."

8. The paper lost much information by restricting to 10,000 most expressed genes and 10,000 most variable CpGs.

Deconvolution of bulk mixtures is unlikely to work in very lowly-variable features. Moreover, in the gene expression analysis, this preprocessing step is also common practice in any downstream analysis. To illustrate how little information there is in the remaining genes, consider genes with at least 3 counts in at least 3 cells, a filtering criteria often applied in single-cell studies. Using these filters, we end up with 11,863 genes for the PBMC dataset; 9,894 of which are included in our simulation. Similarly, for the lung dataset, filtering by at least 5 counts in at least 5 cells, we have 9,735 genes left and 7,920 of which are included in our simulation. In our evaluation of DNA methylation deconvolution, we show the results using both the 10,000 most variable CpGs (Figure 2d,e, and Supplementary Figures S13-S15) and 10,000 randomly selected CpGs (Supplementary Figures S16-19). The latter is expected to reflect the genome-wide performance of different methods.

9. The authors claim: "There are currently no large publicly available bulk datasets with matching cell-type level data for the same group of individuals."

This is not true. Bulk and scRNA-seq data from over 400 ROSMAP donors are available on Synapse.

We agree with the reviewer that this sentence is not accurate, so we removed it.

By “cell-type level data,” we referred to bulk from sorted cells and not single-cell data. RNA-seq and single-cell RNA-seq (scRNA-seq) are inherently different in what they capture, even if coming from the same sample/individual. This has been shown by Velmeshev et al., 2019 Science paper and observed by our group in unpublished data. Of note, the bMIND authors state they resorted to simulations for the exact same reason.

10. Minor: Line 110, "cell-type level means and covariates", "covariates" should be "covariances".

We have revised the text and corrected this typo.

Reviewer #2: This paper presents a method called Unico that is meant to estimate the pure cell type expression matrix given only the bulk expression and cell-type proportions. The authors demonstrate that several of the parameters estimated by Unico are more accurate than competing methods.

The analysis uses a suboptimal "baseline" which the authors define as "naively weighting each bulk profile by the cell-type proportions of the sample" This baseline is uniformly worse than other baselines namely: raw bulk expression and bulk expression with cell type proportions regressed out. These baselines must be included for a proper analysis.

We think the premise of computational deconvolution is twofold:

- (i) Analyzing the data at cell-type resolution may provide insight into cell-type-level variation, and
- (ii) Modeling the heterogeneity of the data better can result in a more robust and accurate analysis. In more detail:

(i) Since a standard analysis of bulk expression (or methylation) levels does not provide a cell-type resolution, bulk and deconvolution estimates cannot be compared on equal ground. In particular, even in the hypothetical scenario where bulk is more correlated (compared to deconvolution estimates) with the cell-type profiles of all cell types, this does not advance our cell-type understanding. Under this criterion, one should think of the deconvolution of a bulk level of a single feature from a single sample as the task of learning the best possible (in terms of correlation, RMSE, etc.) vector of the underlying cell-type levels (which, in general, are unique for each cell type). Bulk levels should not be evaluated for tasks it cannot perform. For that reason, we defined a bulk-based baseline, which intuitively uses the cell-type proportions to map bulk levels to cell-type levels.

(ii) One can evaluate other aspects of deconvolution models, beyond aiming at cell-type-level analysis. Specifically, asking whether deconvolution models are more robust/reliable compared to standard bulk analysis is a fair question that can be evaluated on equal ground. This question can be answered in the context of downstream analysis, such as our B cell analysis and EWAS analysis. As such, all of our downstream analyses use bulk levels as an additional baseline. In all of those cases, the bulk values are accounted for cell-type proportions (as suggested by the reviewer) – in the B cell analysis, this is done by regressing out the cell composition, and in the EWAS analysis, this is done by accounting for them as covariates.

We extensively discuss the above in the Discussion section:

“Our deconvolution of pseudo-bulk RNA mixtures benchmarked the performance of several deconvolution methods. Notably, we did not evaluate the correlation between the cell-type expression and the bulk levels in this analysis. Instead, we showed that Unico is the deconvolution method that provides the most additional cell-type information beyond the bulk levels. At least in some cases, bulk expression is expected

to be more correlated with cell-type expression; in particular, bulk levels are expected to yield an almost perfect correlation for genes presenting the same expression patterns across cell types. Yet, we omit an evaluation of standard bulk levels in this specific analysis and consider them only in downstream analysis. A key reason for that stems from the need to compare bulk and deconvolution estimates on equal ground. This requires dissecting and mapping the variation in bulk levels to different cell types to match the resolution of a cell-type deconvolution output. An arguably natural approach to achieve this is weighting the bulk levels by cell-type proportions, which we included as a baseline model throughout our study.

Another key reason for considering standard bulk levels only in the context of downstream analysis relates to the core motivation of performing deconvolution. The premise of deconvolution is to provide novel biological insights by modeling and differentiating cell-type patterns in the context of conditions. Thus, the utility of deconvolution should ultimately be established by advancing upon bulk-based downstream analysis tasks.”

When we run this analysis using the supplied tutorial code, which we can assume represents the best scenario performance for this method it can be shown that the estimates produced are very similar across cell types in terms of inter-subject variation and are in fact no better than a proportion corrected to the bulk gene expression. This analysis is detailed in the attached PDF.

See our comment on the previous point regarding the comparison to bulk (and adjusted bulk) levels.

The reviewer further points out the high correlation between deconvolution estimates of different cell types in the simulation. This is indeed a characteristic of deconvolution methods. However, we would like to stress that our extensive evaluation provides multiple evidence that Unico’s deconvolution estimates improve upon existing methods. Using simulations, we also show that Unico is the only method that captures substantial cell-type variation beyond what is reflected in bulk measurements (e.g., Figure 2c, Supplementary Figures 1-3 and 5-12).

That said, while a decent portion of our benchmarking directly evaluated the quality of the cell-type deconvolution estimates (i.e., in line with the type of evaluation reported by the reviewer), we strongly believe that the best use of deconvolution methods is not by directly using their cell-type deconvolution estimates, which are merely noisy point estimates. Instead, one should ideally tailor the deconvolution model to each specific downstream analysis task and account for the uncertainty in the underlying tensor.

We extensively discuss this rationale in the Discussion section:

“Evaluating the correlation of deconvolution estimates with cell-type patterns merely represents an indirect (although common) metric for comparing deconvolution methods. Such correlations rely on point estimates of the cell-type patterns, which are expected to be noisy and reflect the ‘‘best guess’’ of a given model. Yet, deconvolution methods such as Unico also model the uncertainty of those point estimates.

This uncertainty can be integrated to perform well-powered downstream statistical analysis without explicitly generating noisy point estimates. As we demonstrate through the analysis of multiple RNA and DNA methylation datasets, this approach advances over a standard bulk analysis in terms of standard performance metrics and its ability to report results at a cell-type resolution.”

Lastly, following the reviewer's comment on shuffling the cell-type labels of the estimated tensor,

we asked if Unico shows an expected performance drop when shuffling the labels of the cell types in our simulations. In this scenario, the deconvolution estimates of each cell type are no longer evaluated with their corresponding ground truth cell-type profiles; for example, estimated NK profiles were evaluated for concordance with the CD4T profiles. We used the same label shuffling used by the reviewer, i.e., $c(2,3,4,5,1)$. We generated a figure similar to Figure 2c, which quantifies the information captured by the deconvolution estimates that cannot be explained by the bulk levels (for information about this quantification, see Methods section on "Deconvolving mixtures of gene expression profiles and estimating cell-type level moments"). We found that 73.5% of the cell-type gene pairs show a decrease in their capacity to capture information that cannot be explained by the bulk levels (Figure R2). This is consistent with a substantial information mismatch/loss as a result of shuffling cell-type labels.

Figure R2. Assessing deconvolution estimates for their information that cannot be explained by pseudo bulk expression across 20 sets of pseudo-bulk mixtures based on PBMC scRNAseq profiles of five cell types (500 samples and 600 randomly selected genes in each set). Boxplots reflect the distribution across genes, the difference between the log-scaled p-values of the effects of the pseudo bulk expression and deconvolution estimates (higher is better; Methods). Shuffling the labels of cell types ("Unico shuffle") performs poorly compared to the non-shuffled evaluation.

The statement "This advancement enhances our capability to conduct powerful large-scale genomic studies at cell-type resolution without the need for cell sorting or single-cell biology." is not supported by the results. The deconvolution problem is clearly not identifiable when using cell proportions and even if reference is included, as is the case with other methods. It is still possible that non-trivial solutions can be obtained in some cases. It would encourage the authors to be more explicit about how and why their method achieves this goal.

Our downstream analysis supports the claim that Unico advances upon existing deconvolution methods. However, in response to the reviewer's criticism, we toned down this sentence, which previously implied that deconvolution may be as good as cell sorting or single-cell measurements. The last sentence of the abstract now reads:

"By deconvolving bulk gene expression and DNA methylation datasets, we demonstrate Unico's superior performance compared to existing methods, enhancing the ability to conduct powerful, large-scale genomic studies at cell-type resolution."

Minor comments

-the benchmarking results from page 8 is using ground truth fraction as input, which in real case you don't have access to. Meanwhile, for 'cibersortx' method it not clear what fraction it is using

We thank the reviewer for raising this practical consideration regarding the quality of the estimated cell-type fractions. We present results under a varying level of noise added to the true cell-type fractions. As expected, the performance of all methods drops as we increase the level of noise. However, Unico performs better than the alternative methods in all scenarios (Supplementary Figures 11 and 12). In particular, we observe that the Unico deconvolution estimates capture cell-type variation that could not be explained by bulk levels in the majority of cell-type gene pairs, as long as the cell-type fraction achieves

≥ 0.7 Pearson correlation with the true cell-type fractions (Supplementary Figures 11 and 12g). Importantly, we would like to emphasize that all the analyses reported in the Results section, except for this particular simulation (deconvolving single-cell data derived pseudo bulk), are based on real data and

did use cell-type proportion estimates obtained from external tools that leverage existing reference data, which are expected to reflect real-world noise better than any simulation.

CIBERSORTx performs the cell-type profile deconvolution in a two-step approach: first using the "cibersortx/fractions" mode to estimate the cell-type proportions, followed by "cibersortx/hires" mode with the estimated proportion file from the previous step passed in by "--cibresults" argument. We thus directly bypassed the first step and provided the prepared ground truth fractions to CIBERSORTx. We clarified this point in the Supplementary Materials:

"Of note, in general, CIBERSORTx estimates cell-type proportions from the bulk input. Here, however, we directly provided it with the ground truth cell-type proportion of the mixtures, as provided to all other methods we benchmarked."

-in page 22 last sentence, they indeed applied non-negativity constraint, but it is for mean only, the deconvolved Z can still contain negative values

The estimated cell-type profiles may include negative values (this is also a property of other deconvolution methods, such as TCA). Empirically, it is less common when working with methylation, where the deviation from the population mean is not as large as sequencing-based data, such as gene expression. In expression-based analyses, we explicitly added a step to address this before computing fold change and other downstream analyses:

"For methods that do not constrain non-negativity in their estimated tensor, for every gene and cell type, we shifted the distribution of the estimates by subtracting the minimum value detected, which enforced non-negativity prior to the \log_{10} transformation. The effect size of a gene that was estimated to have a constant B cell expression level across all samples was set to 0."

-I don't see much difference between methods in Fig.3a

While the effect size distributions of the different methods may not be visually distinct, most of the differences between Unico and the alternative methods are significant. P-values based on a one-sided paired Wilcoxon test are provided in Figure 3a.

-page 31, using cbsx LM22 as signature for fraction deconvolution results can be unreliable, since LM22 is for normal PBMC; if possible should use other targeted reference

We followed the same steps as the authors who first presented this dataset (Newman et al.), and LM22 has since become the most commonly used reference panel in the field. Indeed, for more heterogeneous tissues, especially from tumors, a proper choice of a reference panel could be challenging. Even when using single-cell profiles, samples could also present stronger batch or individual-specific effects than those in normal tissue. That said, for the additional comparison with BayesPrism (see our response to Reviewer 1), we used single-cell FL tumor data as a prior for deconvolution. We observed that the BayesPrism estimated proportions (a byproduct of the deconvolution, which is part of BayesPrism's output) are concordant with CIBERSORTx's estimates derived based on the LM22 panel.

Second round of review

Reviewer 2

Overall while the authors have done a great deal of work addressing the comments I find the results unsatisfactory. From the perspective of a practitioner the most important value of deconvolution is to show that it recovers per-gene inter-sample variation better than trivial competing methods (bulk and bulk cell type regressed). I have not seen evidence of this so far.

The authors have not provided the simple comparison of per-gene correlation improvement over bulk and bulk (cell type regressed). While the author's reasons for the evaluation they picked are well articulated there is no reason to omit this simple baseline. The authors say "The premise of deconvolution is to provide novel biological insights by modeling and differentiating cell-type patterns in the context of conditions." The evaluation I propose gets to the heart of exactly this question.

I would also like to point out that there are many ways to create cell-type resolved datasets that look different from each other and perform well on some metrics but nevertheless provide no information. For example, I could simply keep the original bulk regressed inter subject variance and just adjust the gene mean to match the expected mean for individual cell-types. This approach would pass some tests, for example it would greatly improve sample wise correlation with ground truth, but it would not be a useful way of representing the data as no additional information is actually gained. From the perspective comparing per-gene inter-subject variation is highly informative and arguably the most useful metric.

I also emphasize that this simple comparison is much more intuitive than the various downstream analyses where the multistep nature of the process makes it difficult to assess the specific contribution of deconvolution and designing appropriate baselines becomes challenging.

Previously we had submitted a detailed markdown document with simple demonstrations of some properties of the method. Ideally, the authors would address these directly instead of producing entirely different analyses. However, this would necessitate that the authors agree that gene level correlation with ground truth and their comparisons to bulk and bulk (regressed) are indeed of interest. I find this would be more intuitive than the more complex plots like Figure R2. More generally I would like to see a rebuttal that follows the original evaluation framework we had demonstrated.

I apologize for not spending much time on the mathematical presentation before but upon closer inspection some revision would be helpful.

In the supplement:

-What does "nonlinearly dependent" mean? Please clarify.

-Eq. (6), moving from the 2nd line to the 3rd line, the equality $E[A_{ij} | X_{ij}] = E[A_{ij}]$ (omitting other conditional terms) rely on the independence between A_{ij} and X_{ij} . However, the construction in Eq(8) only gives that A_{ij} and X_{ij} are uncorrelated. In order for the two to be equivalent, you need A_{ij}

and X_{ij} to jointly follow a multivariate normal distribution. This is not stated.

In the main text:

-The main theorem gives the estimator in terms of quantities that are not part of the input. This may confuse readers about what information is assumed to be available and what the estimator depends on.

It would be very helpful if the approach was stated in a more standard statistical learning framework by stating the inputs, objective (purely in terms of the input), and constraints (if any). This would also facilitate comparison to other methods from a conceptual point rather than focusing on the algorithmic optimization details.

Minor issues

Please provide more explanation on the negative values, as well as independent structure in each gene. If negative values are simply shifted by subtracting the minimum value detected (which can be -7000), per gene level correlation indeed not changing, but would modify per sample level correlation. In real application for example if I'm interested in rank-based scores like GSVA, which value should I use.

I would like to see discussion about input sample size requirements. In figure S5-10 there are comparisons between methods with sample size 500/250/100, but not a direct comparison between different sample sizes. So far it looks like we are safe to trust the results up to 100 samples, but how about more samples?

Authors' response to reviewers

We thank the reviewers for their comments and suggestions. We have made a considerable effort to address the remaining comments made by Reviewer 2. Specifically, we:

1. Evaluated Unico and the other deconvolution methods under the evaluation framework suggested by Reviewer 2. Consistent with the conclusions from our previous benchmarking, our results based on the suggested evaluation framework provide strong evidence that Unico performs better than standard analysis of bulk levels and better than competing deconvolution models.
2. Addressed the remainder of the comments and revised the text in the manuscript accordingly.

Below, we provide a point-by-point response to the reviewers' comments (our responses are in blue font). We further provide a version of the main text and supplementary file in which the additional changes made in this revision are annotated in red font, alongside the changes from the previous revision, which are annotated in blue font.

Reviewer #2: Overall while the authors have done a great deal of work addressing the comments I find the results unsatisfactory. From the perspective of a practitioner the most important value of deconvolution is to show that it recovers per-gene inter-sample variation better than trivial competing methods (bulk and bulk cell type regressed). I have not seen evidence of this so far.

We thank the reviewer for their constructive feedback. We agree with the reviewer that it is critical for a deconvolution method to demonstrate its ability to recover per-gene inter-sample variation (hereafter referred to as “inter-sample variation” for short). As we elaborated in our previous revision and emphasized in the manuscript (see second paragraph of the Discussion section), we believe that the main comparison should be with deconvolution methods and other baselines that provide cell-type resolution estimates. Since the output of deconvolution and bulk levels or “corrected bulk” levels (bulk levels with cell-type proportions regressed out) are of different dimensions, we should be more careful with the interpretation and presentation of a direct comparison between them.

That said, we agree with the reviewer that comparing Unico and the other deconvolution methods with bulk and corrected bulk as baselines may offer additional insight into how well each method captures inter-sample variation. Therefore, as detailed in the remainder of this rebuttal, we added additional analysis and supplemental figures to address the reviewer's concern. Specifically, we followed the exact evaluation framework outlined in the markdown file provided by the Reviewer, which includes comparison of deconvolution with bulk and corrected bulk levels, as well as other evaluations. We note that the tutorial dataset used by the reviewer corresponds to the first batch of the simulation dataset (out of 20 we used in our benchmarking), which was derived from PBMC scRNA-seq profiles from Stephenson et al. We have since extended our analysis to also include 20 batches of simulation datasets derived from lung scRNA-seq profiles.

Please find attached the Rebuttal Supplementary File **Reviewer2-evaluation-Unico.pdf**, which follows the evaluation framework proposed by this reviewer. In addition to reconstructing the same step-by-step evaluation by the reviewer (see Rebuttal Supplementary File **Reviewer2-Unico-code-review.pdf**), we printed additional information that quantifies the aspects the reviewer suggested as important from a practitioner's perspective. In addition, we provide the same analysis for each competing method benchmarked in a separate notebook file (e.g., see Supplementary Rebuttal File **Reviewer2-evaluation-TCA.pdf** for the evaluation of TCA). We further included a direct comparison of each method to Unico at the end of its corresponding notebook. For the reviewer's convenience and ease of reproducibility, we saved the deconvolution results of the competing methods in an RDS file (**Z.hat.list.RDS, also available via Google drive**).

https://drive.google.com/file/d/1ybe6ouls4djmvl7vyL7E_YkBKif8eDvV/view?usp=share_link

The authors have not provided the simple comparison of per-gene correlation improvement over bulk and bulk (cell type regressed). While the author's reasons for the evaluation they picked are well articulated there is no reason to omit this simple baseline. The authors say

“The premise of deconvolution is to provide novel biological insights by modeling and differentiating cell-type patterns in the context of conditions.”

Under the evaluation framework suggested by the reviewer, we found that

(1) **Comparison with bulk levels:** The Unico deconvolution estimates of all cell types are more correlated with the true underlying cell-type-specific profiles compared to a naive baseline of bulk levels. In particular, the fraction of genes for which Unico performs better than bulk is 0.928, 0.898, 0.865, 0.917, and 0.922 for CD4 T cells, NK, CD8 T cells, monocytes, and B cells, respectively. Furthermore, Unico is ranked as the top-performing method (Rebuttal Figures R3a,c and R4). For example, the second-best method (TCA), also improves upon the bulk baseline across the majority (>50%) of genes, but only within a subset of the cell types (66.8%, 52.2%, and 57.7% of the genes for CD4 T cells, NK, and monocytes respectively). The full results for all methods are provided in the analysis notebooks.

Figure R3. Presented are the fraction of genes for which the ground truth cell-type profiles are more correlated with the deconvolution estimates than with (a) the bulk levels and (b) the corrected bulk levels. Barplots and error bars represent means and one standard deviation across 20 sets of pseudo-bulk mixtures from PBMC scRNAseq profiles (500 samples and 600 genes in each set). (c-d) Similar to (a-b), only using pseudo-bulk mixtures from lung scRNAseq profiles of four cell types. Results are based on the evaluation framework suggested by the reviewer.

Figure R4. Scatter plots represent the correlation between each gene's ground truth cell-type profile and its deconvolution estimates (y-axis), relative to the correlation between the ground truth profile and the **bulk baseline** (x-axis). Each row corresponds to pairwise comparisons between different methods: (a) TCA, (b) bMIND, (c) BayesPrism, and (d) CIBERSORTx, against Unico. Dots falling below the diagonal line indicate worse performance than the baseline. Results are based on the evaluation framework suggested by the reviewer.

(2) **Comparison with corrected bulk levels:** The Unico deconvolution estimates of all cell types are also more correlated with the true underlying cell-type-specific profiles compared to “corrected bulk” levels. In particular, the fraction of genes for which Unico performs better than corrected bulk is 0.722, 0.642, 0.520, 0.828, and 0.683 for CD4 T cells, NK, CD8 T cells, monocytes, and B cells, respectively. Similar to (1), Unico is ranked as the top-performing method (Rebuttal Figures R3b,d and R5). The second-best method (TCA) does not improve upon a correct bulk baseline for most genes (Rebuttal Figures R3b,d and R5a). The full results for all methods are provided in the analysis notebooks.

Figure R5. Scatter plots represent the correlation between each gene's ground truth cell-type profile and its deconvolution estimates (y-axis), relative to the correlation between the ground truth profile and the **corrected bulk baseline** (i.e., bulk levels with cell-type proportions regressed out; x-axis). Each row corresponds to pairwise comparisons between different methods: (a) TCA, (b) bMIND, (c) BayesPrism, and (d) CIBERSORTx, against Unico. Dots falling below the diagonal line indicate worse performance than the baseline. Results are based on the evaluation framework suggested by the reviewer.

(3) **Evaluation of cell-type specificity:** The reviewer suggested in their evaluation framework that the Unico deconvolution estimates have low cell-type specificity. This was demonstrated by the reviewer in two ways: first, by showing a histogram that presented an inflated correlation between the estimates of different cell types, and second, by showing a high correlation between the Unico estimates and the ground-truth cell-type levels, even when the order of cell types in the deconvolution estimates was randomly shuffled.

Following the evaluation framework suggested by the reviewer, we shuffled the cell types in the Unico deconvolution estimates and asked whether they correlate with the ground truth cell-type profiles similarly well as the unshuffled estimates. We found that the unshuffled Unico estimates perform better in terms of correlation than the shuffled estimates, indicating sensitivity to the shuffling (and, therefore, cell-type specificity). Particularly, the fraction of genes for which the unshuffled estimates performed better than the shuffled estimates is 0.798, 0.607, 0.840, 0.897, and 0.643 for CD4 T cells, NK, CD8 T cells, monocytes, and B cells, respectively. These results are consistent with the results from our previous revision round, which also showed cell-type specificity using cell-type shuffling (Figure R2 from the previous revision).

Notably, our results indicate an increased sensitivity to shuffling of a cell type from the lymphoid compartment with a cell type from the myeloid lineage (e.g., monocytes paired with B cells; the fraction of genes for which shuffled is better than unshuffled is 0.897) compared to shuffling within the same cell-type lineage (e.g., B cells paired with CD4 T cells; fraction 0.643). This observation aligns with our expectation that gene expression profiles are more coordinated among cell types that are proximal on the cell-type lineage differentiation tree (Supplementary Material, pages 43-44, lines 42-49).

Finally, we incorporated the new analysis and suggested visualization by the reviewer comparing Unico and the other deconvolution methods with the bulk and corrected bulk levels into the manuscript. While we used standard linear correlation in the rebuttal figures (i.e., the original code used by the reviewer), for consistency with all of the analysis throughout our manuscript, we used robust correlation, among other measures to alleviate biases in our benchmarking (see "Calculating robust linear correlation" in the Methods section, Supplementary figure S29 and the last paragraph in Supplementary Methods "Generating pseudo-bulk mixtures from scRNAseq profiles" for justification).

We added the following text to the Discussion section, referring to the new additional figures: *"However, for completeness, we extended our benchmarking to compare Unico with bulk levels and confirmed that Unico's estimates are more correlated with the true underlying cell-type expression profiles (Supplementary Figures S35-S38)."*

I would also like to point out that there are many ways to create cell-type resolved datasets that look different from each other and perform well on some metrics but nevertheless provide no information. For example, I could simply keep the original bulk regressed inter subject variance and just adjust the gene mean to match the expected mean for individual cell-types. This approach would pass some tests, for example it would greatly improve sample wise correlation with ground truth, but it would not be a useful way of representing the data as no additional information is actually gained. From the perspective comparing per-gene inter-subject variation is highly informative and arguably the most useful metric.

I also emphasize that this simple comparison is much more intuitive than the various downstream analyses where the multistep nature of the process makes it difficult to assess the specific contribution of deconvolution and designing appropriate baselines becomes challenging.

We completely agree with the reviewer that comparing inter-subject variation, stratified per gene, is crucial. For this reason, we have extensively used multiple metrics to evaluate the inter-subject variation of the deconvolution estimates throughout our manuscript (in addition to the evaluation framework proposed by the reviewer). In contrast, and in line with the rationale and counterexample proposed by the reviewer, several studies in the literature report additional cross-gene metrics, such as per-sample correlation across genes. As implied by the reviewer, such metrics only quantify “global” cell-type patterns and offer little to no insight into inter-subject variation. For this reason, we avoid using such metrics.

We further agree with the reviewer that directly evaluating inter-subject variation is more intuitive than comparing methods based on their performance in downstream analysis tasks. However, we would like to reiterate the usefulness of the latter. In particular, as we elaborate in the third paragraph of our Discussion section, evaluating the inter-subject variation of deconvolution estimates relies on point estimates, which are expected to be noisy and reflect the “best guess” of a given model. Yet, deconvolution methods such as Unico also model the uncertainty of those point estimates. In theory, properly combining uncertainty estimates in downstream analyses is expected to improve performance. Therefore, we believe that a comprehensive evaluation of deconvolution methods should combine benchmarking of downstream analysis tasks.

Moreover, we believe that evaluating the effect of deconvolution on downstream analysis is important for evaluating the additional information provided by deconvolution that is not captured by bulk levels. Concretely, under a simple evaluation of inter-subject variation, it is difficult to interpret whether an improvement of deconvolution over a naive bulk (or corrected bulk) baseline is meaningful in terms of its impact on downstream applications. This stems from

the scale of the metrics defined to evaluate the inter-subject variation. It is a priori unclear whether a small or large improvement in these metrics translates to improvement in the downstream application. Our evaluation based on downstream analysis suggests that Unico's seemingly modest performance improvement over bulk in explaining inter-subject variation (Rebuttal Figures R3-R5), translates into a substantial improvement in downstream analyses of interest to practitioners.

For example, in our evaluation of cell-type-level EWAS, we observe relatively poor consistency across studies for standard bulk analysis (mean MCC = 0.45 and 0.62 for sex and age, respectively; despite accounting for cell-type proportions) compared to a substantial increase in consistency when using Unico (MCC = 0.94 and 0.87 for sex and age respectively).

Previously we had submitted a detailed markdown document with simple demonstrations of some properties of the method. Ideally, the authors would address these directly instead of producing entirely different analyses. However, this would necessitate that the authors agree that gene level correlation with ground truth and their comparisons to bulk and bulk (regressed) are indeed of interest. I find this would be more intuitive than the more complex plots like Figure R2. More generally I would like to see a rebuttal that follows the original evaluation framework we had demonstrated.

We are grateful to the reviewer for recommending additional evaluation metrics that were missing from our earlier benchmarking. As we described in detail above, we followed the evaluation framework suggested in the markdown document by the reviewer. The new set of results complements our proposed analysis in the previous revision round (Figure R2) and existing results presented in the manuscript (Figure 2 and Supplementary Figures S1-12). Altogether, they support a consistent narrative that Unico is the most effective and best-performing deconvolution method, providing the most information beyond baseline bulk (or corrected bulk) levels. We believe our results adequately address the concerns raised by the reviewer.

I apologize for not spending much time on the mathematical presentation before but upon closer inspection some revision would be helpful.

In the supplement:

-What does “nonlinearly dependent” mean? Please clarify.

-Eq. (6), moving from the 2nd line to the 3rd line, the equality $E[A_{ij} | X_{ij}] = E[A_{ij}]$ (omitting other conditional terms) rely on the independence between A_{ij} and X_{ij} . However, the construction in Eq(8) only gives that A_{ij} and X_{ij} are uncorrelated. In order for the two to be equivalent, you need A_{ij} and X_{ij} to jointly follow a multivariate normal distribution. This is not stated.

We thank the reviewer for pointing out this oversight on our end. Indeed, our derivation requires assuming that zero covariance indicates independence, which is not generally true. The reviewer correctly points out that if A_{ij} and X_{ij} jointly follow a multivariate normal distribution, then this assumption is satisfied. We added an additional assumption to remedy that, now stating that X_{ij} is assumed to follow a joint distribution, which allows us to construct independence between A_{ij} and X_{ij} .

In addition to updating the claim in Theorem, we now include the following in the main text to acknowledge this limitation (under “Estimating the underlying 3D tensor with Unico”):

“The assumption that bulk levels follow a normal distribution limits the generalizability of this theoretical result. However, it allows us to analytically derive an efficient estimator. Importantly, our benchmarking results suggest that empirically, this estimator provides a good approximation even under empirical violations of the assumption (e.g., in deconvolving gene expression levels). Furthermore, this assumption does not affect the distribution-free estimation of the model parameters, which we describe in the following subsection.”

We further revised the Supplementary Methods to reflect the new assumption. We removed the ambiguous wording on “nonlinearly dependent” and clarified the additional assumption needed for this condition to hold (Supplementary Methods, pages 45-46, lines 91-104).

Subsection S2.2 in the Supplementary Methods now includes:

“Assuming we can construct A_{ij} such that if A_{ij} , X_{ij} are uncorrelated, then they are independent, then given $X_{ij} = x_{ij}$, we can express Z_{ij} using the relation $A_{ij} - B_{ij} X_{ij}$ and calculate the conditional expectation as follows

X_{ij} is assumed to follow a normal distribution, which means A_{ij} , a linear transformation of X_{ij} , also follows a normal distribution. Since X_{ij} , A_{ij} jointly follow a multivariate normal distribution, Equation (7) can hold if A_{ij} and X_{ij} are uncorrelated, which would imply independence. Thus, we can satisfy Equation (7) by requiring the cross-covariance vector between A_{ij} and X_{ij} to be zero given $\{\theta_j, w_i, c^{(1)}_i, c^{(2)}_i\}$ ”.

In the main text:

-The main theorem gives the estimator in terms of quantities that are not part of the input. This may confuse readers about what information is assumed to be available and what the estimator depends on.

We thank the reviewer for this feedback. To improve clarity, we reordered some of our previous explanations to avoid confusion about the information assumed to be available for the Unico estimator. Specifically, we incorporated the following paragraph right before stating Theorem 1 (see "Estimating the underlying 3D tensor with Unico" in the Method section, page 24, pages 308-313):

“The following theorem provides an analytical solution for the 3D tensor estimator \hat{z}_{ij} under the

Unico model in Equations(3)-(7), given the model parameters $\{\theta_{ij}\}$ and model input: bulk profiles

$\{x_{ij}\}$, cell-type proportions $\{w_{ij}\}$, and cell-type- and tissue-level covariates $\{c_i^{(1)}, \dots, c_i^{(2)}\}$. In practice, as mentioned above, cell-type proportions estimated using external decomposition methods are provided as part of the input, from which we subsequently estimate the model parameters, as described later.”

It would be very helpful if the approach was stated in a more standard statistical learning framework by stating the inputs, objective (purely in terms of the input), and constraints (if any). This would also facilitate comparison to other methods from a conceptual point rather than focusing on the algorithmic optimization details.

We thank the reviewer for suggesting we present the method in a more standard statistical learning framework in the manuscript. We have now

- (1) expanded and incorporated an algorithmic overview in the Method section (pages 26-27, lines 346-373) that introduces readers to Unico's non-parametric GMM estimation approach.
- (2) clearly stated the inputs required, the core objective (matching the first and second moment conditions with appropriate weights), and the model parameters being estimated in each round of GMM update.
- (3) specified the non-technical constraints (non-negative constraints on the population-level cell-type estimates)

We deferred more technical details, such as the construction of moment condition weights in each iteration and constraints like symmetric and positive semi-definiteness of the covariance matrix, to the Supplementary Methods. We believe this addition provides more clarity regarding the conceptual difference from other methods. We further provide a discussion on the latter (see Subsection S2.1.2 “Unico in the context of previous deconvolution methods” in the Supplementary Methods).

Minor issues

Please provide more explanation on the negative values, as well as independent structure in each gene. If negative values are simply shifted by subtracting the minimum value detected (which can be -7000), per gene level correlation indeed not changing, but would modify per sample level correlation. In real application for example if I'm interested in rank-based scores like GSVA, which value should I use.

We thank the reviewer for pointing this out. The possibility of negative values is inherent in our framework since Unico's deconvolution estimator is not bound to non-negative values. Extreme negative values can be explained by outlier input values. There are three types of inputs, each of which may lead to extreme deconvolution estimates:

1. Covariates: Since the estimated effect sizes are essentially mean effect sizes, combining them with extreme covariate values may result in extreme deconvolution estimates, either positive or negative.
2. Cell-type proportions: Similarly, a very unusual cell-type proportions profile may lead to outlier deconvolution estimates.
3. Bulk levels: Similarly, outlier bulk levels may lead to outlier deconvolution estimates

In the tutorial dataset used by the reviewer (the first batch of simulation dataset derived from PBMC scRNAseq data from Stephenson et al.), the reason for the extreme negative estimate is point (3). We provide a Supplementary Rebuttal File showing that (**Reviewer2-large-negative.pdf**). Briefly, we observe that the most extreme negative estimates of Unico all stem from outlier bulk levels. In other words, if Unico estimates an extreme value for a specific gene of a specific sample (in some cell type), then that sample is likely to have an extreme bulk level in that gene (note that the parameters of different genes are estimated independently). In particular, the extremely negative estimate of Unico pointed out by the reviewer (< -7000) corresponds to a sample exhibiting an extreme bulk level expression of the estimated gene ($X_{ij} = 9723$, compared to a median of 9, and more than 16 standard deviations away from the mean; see **Reviewer2-large-negative.pdf**).

We find that empirically, there are not many negative values (e.g., only ~1% of the estimates are smaller than -1 in our analysis; see **Reviewer2-evaluation-Unico.pdf**). However, extreme negative values may affect downstream analysis. Therefore, one should handle them similarly to how they would treat outlier data points in any other application. For example, to alleviate outlier effects on sample-level correlations across multiple genes (i.e., the case considered by the reviewer), one may want to consider different solutions, including but not limited to replacing outlier values with missing values, standardizing the distribution of each gene (i.e., standard deviation of 1 across the deconvolution estimates of all samples), or excluding samples with outlier values (either at the bulk or deconvolution level).

We revised our manuscript to include information about the potential effect of outliers in the input. The Subsection titled "Implementation of Unico and practical considerations" under the Methods section now includes:

“Lastly, the Unico deconvolution estimator can occasionally yield extreme values — either positive or negative — which is an inherent property of the framework, as it does not impose non-negativity constraints. In particular, extreme negative estimates often arise from outlier input values. Because the estimator relies on bulk-level data, outliers in the bulk measurements can propagate and lead to extreme deconvolution outputs. Similarly, since the estimated effect sizes represent averages across individuals, applying them to extreme covariate values can amplify the resulting estimates. As extreme values may influence downstream analyses, we recommend handling such values using standard practices, consistent with how one would treat outlier data points in other analytical settings”.

I would like to see discussion about input sample size requirements. In figure S5-10 there are comparisons between methods with sample size 500/250/100, but not a direct comparison between different sample sizes. So far it looks like we are safe to trust the results up to 100 samples, but how about more samples?

We revisited our benchmarking and considered additional scenarios with various sample sizes. Specifically, we now consider the following range of sample sizes: 1000, 500, 250, 100, 50 and include direct comparison to better visualize the performance trend as a function of decreasing sample sizes (Supplementary Figures S13 and S14). Our results suggest that an effective deconvolution requires at least 100 samples (to capture cell-type information beyond bulk levels), with improvement plateauing above 250 samples (based on inter-sample correlation).

We included a discussion about these new results in the Results section (under the Subsection titled “Establishing a new state-of-the-art deconvolution for bulk genomics”). Specifically, the end of the fourth paragraph now reads (pages 11-12, lines 124-133):

Comparing Unico across increasing sample sizes demonstrates an expected increase in correlation between the deconvolution estimates and the true underlying cell-type-specific expression levels, with improvement plateauing above 250 samples (Supplementary Figures S13 and S14)”.

and the fifth paragraph now reads:

“We asked whether the 3D tensors estimated by Unico and other methods explain the variation of the true tensor beyond the pseudo-bulk input... Evaluating Unico based on this metric across various sample sizes, renders Unico's deconvolution effective for datasets greater than 100 samples (Supplementary Figures S13 and S14)”.

Third round of review

Reviewer 2

1. I appreciate the authors' agreement that the bulk-regressed baseline provides a more appropriate and informative comparison than the original baseline, which simply weights each bulk profile by the sample's cell-type proportions. However, based on the text and the results presented in Figure 2b, it appears that the original, weaker baseline is still being used as the primary point of comparison. This choice likely inflates the perceived performance of the method and may mislead readers about the true magnitude of improvement. Given that Unico shows modest but consistent gains over the bulk-regressed baseline, I believe those results offer a strong and honest foundation for publication. I urge the authors to reflect this more appropriate comparison in the main figures and text.

Comparisons to the strong baseline as produced by the authors

Comparisons in the paper

2. Additionally, I noticed the newly added statement:

“Yet, deconvolution methods such as Unico also model the uncertainty of those point estimates. This uncertainty can be integrated to perform well-powered downstream statistical analysis without explicitly generating noisy point estimates.”

This is the only mention of uncertainty in the manuscript, and it is not supported by any corresponding results, methodological description, or guidance on how the uncertainty is estimated or used. If uncertainty modeling is a substantive feature of the method, I would encourage the authors to elaborate and provide evidence.

3. Finally, I still find the mathematical formulation difficult to follow. While the authors have made substantial changes, the introduction of additional variables has, from my perspective, made the presentation more opaque. I have written up some notes (see attached separate pdf) that I believe present the method more clearly by:

- Explicitly listing all the variables and dimensions and specifying the main GLM form
- Minimizing the number of introduced variables to keep the core idea focused
- Rewriting the posterior inference using standard operations
- Deferring covariates to a later extension rather than including them in the base formulation

I realize there may be benefits to both presentations but I urge the authors to consider including (even as a supplementary note) the simplified version to allow readers from a broader set of backgrounds to appreciate the methodological innovation.

Authors' response to reviewers

Here, we provide a short point-by-point response to the second reviewer's remaining comments (provided to us in a pdf file). We edited the manuscript to address these comments (edits are marked with green font in the manuscript).

1. Corrected bulk as a stronger baseline:

We appreciate the reviewer for their thoughtful feedback and are pleased that our revisions conveyed that our results “offer a strong and honest foundation for publication”. In response to this comment suggesting we highlight the evaluation of a bulk (and “bulk-regressed”) baseline, we have added a new paragraph to the Results section (fifth paragraph under “Establishing a new state-of-the-art deconvolution for bulk genomics”), reporting a direct comparison of Unico and the other deconvolution methods with bulk expression (and “bulk-regressed” expression, i.e., cell-type corrected bulk expression). We believe that highlighting these results in the main Results section (including in Figure 2c) and referring to the results of a comprehensive comparison of all deconvolution methods with bulk expression across all datasets and evaluation settings (Supplementary Figures S1-S3 and S5-S18) addresses the concerns of the reviewer about misleading readers about the magnitude of improvement. Since comparing bulk levels with deconvolution methods is not strictly equivalent—

given that bulk profiles do not resolve cell-type-specific levels—we added an explicit clarification in the new paragraph to prevent possible misinterpretation of our results.

2. Clarification on "uncertainty":

Our discussion on uncertainty meant to clarify that point estimates of deconvolution models are expected to be noisy, and that deconvolution models can quantify this noise (i.e., uncertainty) and account for it in downstream analysis. This is a general point of discussion on deconvolution rather than a specific one about Unico. We revised the statement for clarity and elaborated to avoid misinterpretation due to potentially ambiguous terminology (see the second paragraph of the Discussion section).

3. Simplify methodology formulation:

We thank the reviewer for raising this concern and for the concrete suggestion regarding how to reformulate the presentation of the algorithm. We share the concern that the dense mathematical formulation may come across as challenging to some readers. In light of this, we have made a serious effort to systematically simplify and clarify the mathematical notation throughout the Methods section. We believe these edits address the reviewer's suggestion to "allow readers from a broader set of backgrounds to appreciate the methodological innovation".

Concretely, we have adopted the reviewer's suggestion to explicitly state the dimensions of variables wherever possible. We also opted for a vectorized notation on key variables to avoid excessive subscripts. Although cell-type and tissue-level covariates are critical in practical applications, we agree with the reviewer that their inclusion in the main text could introduce unnecessary complexity, potentially obscuring the core methodological innovation. Therefore, we followed with the reviewer's suggestion to defer their introduction to the Supplementary Methods. This change also reduces the number of variables presented in the main Methods section.

We respectfully disagree with the suggestion to frame the entire model under Gaussian distribution, which the reviewer refers to as "standard operation". The generative model underlying Unico, described in Equations (3)-(6), makes no assumptions on the distribution of the underlying components of variation, and the parameter estimation is performed under the generalized method of moments framework, which makes no assumption on the underlying distribution. While some of our theoretical results rely on the assumption that cell-type levels follow Gaussian distributions (Theorem 1), describing the entire framework under a Gaussian model would be incorrect and misleading.